

# A scientific algorithm to simultaneously retrieve carbon monoxide and methane from TROPOMI onboard Sentinel-5 Precursor

Oliver Schneising[1], Michael Buchwitz[1], Maximilian Reuter[1], Heinrich Bovensmann[1], John P. Burrows[1], Tobias Borsdorff[2], Nicholas M. Deutscher[3], Dietrich G. Feist[4,5], David W. T. Griffith[3], Frank Hase[6], Christian Hermans[7], Laura T. Iraci[8], Rigel Kivi[9], Jochen Landgraf[2], Isamu Morino[10], Justus Notholt[1], Christof Petri[1], David F. Pollard[11], Sébastien Roche[12], Kei Shiomi[13], Kimberly Strong[12], Ralf Sussmann[14], Voltaire A. Velazco[3], Thorsten Warneke[1], and Debra Wunch[12]

[1]Institute of Environmental Physics (IUP), University of Bremen FB1, Bremen, Germany
[2]SRON Netherlands Institute for Space Research, Earth Science Group (ESG), Utrecht, the Netherlands
[3]Centre for Atmospheric Chemistry, School of Earth, Atmosphere and Life Sciences, University of Wollongong, Wollongong, Australia
[4]Deutsches Zentrum für Luft- und Raumfahrt, Institut für Physik der Atmosphäre, Oberpfaffenhofen, Germany
[5]Max Planck Institute for Biogeochemistry, Jena, Germany
[6]Karlsruhe Institute of Technology (KIT), Institute for Meteorology and Climate Research (IMK-ASF), Karlsruhe, Germany
[7]Royal Belgian Institute for Space Aeronomy, Brussels, Belgium
[8]Atmospheric Science Branch, NASA Ames Research Center, Moffett Field, USA
[9]Finnish Meteorological Institute, Space and Earth Observation Centre, Sodankylä, Finland
[10]Satellite Remote Sensing Section and Satellite Observation Center, Center for Global Environmental Research, National Institute for Environmental Studies (NIES), Tsukuba, Japan
[11]National Institute of Water and Atmospheric Research (NIWA), Lauder, New Zealand
[12]Department of Physics, University of Toronto, Toronto, Canada
[13]Japan Aerospace Exploration Agency (JAXA), Tsukuba, Japan
[14]Karlsruhe Institute of Technology (KIT), Institute for Meteorology and Climate Research (IMK-IFU), Garmisch-Partenkirchen, Germany

**Correspondence:** O. Schneising (oliver.schneising@iup.physik.uni-bremen.de)

**Abstract.** Carbon monoxide (CO) is an important atmospheric constituent affecting air quality and methane ($CH_4$) is the second most important greenhouse gas contributing to human-induced climate change. Detailed and continuous observations of these gases are necessary to better assess their impact on climate and atmospheric pollution. While surface and airborne measurements are able to accurately determine atmospheric abundances on local scales, global coverage can only be achieved
5    using satellite instruments.

The TROPOspheric Monitoring Instrument (TROPOMI) onboard the Sentinel-5 Precursor satellite, which was successfully launched in October 2017, is a spaceborne nadir viewing imaging spectrometer measuring solar radiation reflected by the Earth in a push-broom configuration. It has a wide swath on the terrestrial surface and covers wavelength bands between the ultraviolet (UV) and the shortwave infrared (SWIR) combining a high spatial resolution with daily global coverage. These
10   characteristics enable the determination of both gases with unprecedented level of detail on a global scale introducing new areas of application.



Abundances of the atmospheric column-averaged dry air mole fractions XCO and XCH$_4$ are simultaneously retrieved from TROPOMI's radiance measurements in the $2.3\,\mu\text{m}$ spectral range of the SWIR part of the solar spectrum using the scientific retrieval algorithm Weighting Function Modified DOAS (WFM-DOAS). We introduce the algorithm in detail, including expected error characteristics based on synthetic data, a machine learning-based quality filter and a shallow learning calibration procedure applied in the post-processing of the XCH$_4$ data. The quality of the results based on real TROPOMI data is assessed by validation with ground-based Fourier Transform Spectrometer (FTS) measurements providing realistic error estimates of the satellite data: The XCO data set is characterised by a random error of $5.1\,\text{ppb}$ ($5.7\%$) and a systematic error of $1.9\,\text{ppb}$ ($2.1\%$); the XCH$_4$ data set exhibits a random error of $14.0\,\text{ppb}$ ($0.8\%$) and a systematic error of $4.4\,\text{ppb}$ ($0.2\%$). The natural XCO and XCH$_4$ variations are well captured by the satellite retrievals, which is demonstrated by a high correlation to the reference data ($R = 0.97$ for XCO and $R = 0.91$ for XCH$_4$ based on daily averages).

We also present selected results from mission start until end of 2018, including a first comparison to the operational products and examples of the detection of emission sources in a single satellite overpass, such as CO emissions from the steel industry and CH$_4$ emissions from the energy sector.

## 1 Introduction

Carbon monoxide (CO) is an atmospheric pollutant compromising air quality. It is a colourless, odorless, and tasteless gas, that can disrupt the transport of oxygen by hemoglobin in the red blood cells after inhalation of high doses, thus having the ability to cause severe health problems (Omaye, 2002). Its lifetime of about one to two months allows the usage as tracer of long-range transport of pollution. CO plays a central role in tropospheric chemistry acting as a precursor to tropospheric ozone (The Royal Society, 2008), which is another pollutant considered harmful to public health and a greenhouse gas. Moreover, it is the largest direct sink of the hydroxyl radical (OH) affecting the self-cleansing capacity of the atmosphere, as the consumed OH cannot deplete other atmospheric constituents such as methane anymore. Hence, CO can be interpreted as an indirect agent of climate change, because it is affecting concentrations of direct greenhouse gases.

Methane (CH$_4$) is an important long-lived anthropogenically released greenhouse gas. It is second only to carbon dioxide (CO$_2$), which accounts for the largest share of radiative forcing caused by human activities since 1750. CH$_4$ is less abundant in the atmosphere than CO$_2$, but it has a considerably higher global warming potential per unit mass. An accurate understanding of the sources and sinks of CH$_4$ is indispensable to reliably predict future climate. Due to its relatively long atmospheric perturbation lifetime (budget lifetime multiplied by feedback factor) of about 12 years (Prather et al., 2012; Holmes et al., 2013)), CH$_4$ is well-mixed in the atmosphere and the signals in question are typically only small variations on top of large background concentrations. Therefore, the requirements on the precision and accuracy of atmospheric CH$_4$ measurements are demanding (Meirink et al., 2006; Bergamaschi et al., 2009; World Meteorological Organization, 2006, 2011).

Detailed and continuous observations with global coverage of both gases are needed to improve our understanding of the climate system, tropospheric chemistry, and atmospheric transport processes. This objective can only be achieved using satellite instruments. Several spaceborne instruments have been measuring CO and CH$_4$ on a global scale up to now including the





**Table 1.** Summary of the TROPOMI NIR and SWIR spectral bands and their key features (Rozemeijer and Kleipool, 2018).

| Spectrometer | NIR | | SWIR | |
|---|---|---|---|---|
| Band ID | 5 | 6 | 7 | 8 |
| Spectral Range [nm] | $661 - 725$ | $725 - 786$ | $2300 - 2343$ | $2343 - 2389$ |
| Spectral Resolution FWHM [nm] | $0.34 - 0.35$ | | 0.227 | 0.225 |
| Spectral Sampling [nm] | 0.126 | | 0.094 | |
| Spatial Sampling [km$^2$] | $7.1 \times 3.6$ | | $7.1 \times 7.5$ | |
| Detector Binning Factor | 2 | | 1 | |

Atmospheric Infrared Sounder (AIRS) (McMillan et al., 2005; Xiong et al., 2008), the Tropospheric Emission Spectrometer (TES) (Luo et al., 2015; Worden et al., 2012) and the Infrared Atmospheric Sounding Interferometer (IASI) (Clerbaux et al., 2009), which observe emissions in the thermal infrared (TIR) and are mainly sensitive to mid/upper-tropospheric abundances. For CO this category is expanded by the Measurement of Pollution in the Troposphere (MOPITT) instrument (Drummond et al., 2010), which combines observations of spectral features in the TIR and in the shortwave infrared (SWIR) increasing surface-level sensitivity in some scenes (Worden et al., 2010).

Nearly equal sensitivity to all altitude levels including the boundary layer can be achieved from radiance measurements of reflected solar radiation in the SWIR part of the spectrum. This was first demonstrated by retrievals from the SCanning Imaging Absorption spectroMeter for Atmospheric CHartographY (SCIAMACHY) instrument (Burrows et al., 1995; Bovensmann et al., 1999) onboard ENVISAT for CO (Buchwitz et al., 2004; de Laat et al., 2010) and CH$_4$ (Buchwitz et al., 2005; Frankenberg et al., 2006) in the $2.3\,\mu m$ or $1.6\,\mu m$ spectral range. ENVISAT was launched in 2002 and the end of mission was declared after ten years in orbit due to unexpected loss of contact with the satellite in 2012. The Thermal And Near infrared Sensor for carbon Observations Fourier Transform Spectrometer (TANSO-FTS) onboard the Greenhouse gases Observing SATellite (GOSAT) (Kuze et al., 2009), which was launched in 2009, also yields atmospheric CH$_4$ with high near-surface sensitivity but with a fairly sparse spatial sampling interval of about $160\,km$ in five-point across track mode between its $10\,km$ diameter circular footprints. Its successor GOSAT-2 (launched in 2018) has an extended spectral range and is designed to additionally measure CO.

The launch of the Sentinel-5 Precursor (Sentinel-5P) satellite in October 2017 with the TROPOspheric Monitoring Instrument (TROPOMI) onboard (Veefkind et al., 2012) can be considered a game changer for the determination of atmospheric composition from space. TROPOMI is a spaceborne nadir viewing imaging spectrometer measuring solar radiation reflected by the Earth in a push-broom configuration. It has a swath width of $2600\,km$ and allows the analysis of several atmospheric species with unprecedented level of detail by combining high precision and spatial resolution with daily global coverage. TROPOMI measures radiances between the ultraviolet (UV) and the shortwave infrared (SWIR) in 8 bands. The characteristics of the TROPOMI NIR and SWIR bands are summarised in Table 1.





CO and CH$_4$ can be retrieved from radiance measurements in TROPOMI's SWIR bands. For these bands, the spatial resolution of the nadir measurements is typically $7 \times 7\,\mathrm{km}^2$, which is almost 40 times finer than for SCIAMACHY. In contrast to TANSO, the imaging capabilities of TROPOMI provide three orders of magnitude more measurements without gaps, thus facilitating real global maps of CO and CH$_4$ in a short time. The unique combination of high precision, spatiotemporal res-

olution, and coverage enables new fields of application. As large sources are readily detected in a single overpass, emission monitoring and air quality assessments are only two examples of the new prospects TROPOMI offers. First applications concerning CO and CH$_4$ have already been highlighted and demonstrated in recent publications (Hu et al., 2018; Borsdorff et al., 2018, 2019; Schneising et al., 2019).

As in the fields of weather and climate modelling, ensemble approaches have recently acquired an increased importance

in the context of satellite observations, aiming at benefitting from a larger range of possible realisations of different physical aspects (Reuter et al., 2013) or to analyse to what extent specific geophysical findings depend on particular characteristics of an algorithm or instrument (Buchwitz et al., 2017). Along these lines, it is worthwhile to have a set of distinct retrieval algorithms for each analysed atmospheric constituent at hand. Here we introduce a scientific algorithm to retrieve CO and CH$_4$ simultaneously from TROPOMI, which differs from the operational algorithms in several respects (Landgraf et al., 2016;

Hu et al., 2016) (see also Section 4.1).

## 2   WFM-DOAS retrieval algorithm

The Weighting Function Modified Differential Optical Absorption Spectroscopy (WFM-DOAS) algorithm (Buchwitz et al., 2006, 2007; Schneising et al., 2011, 2014) is a least-squares method based on scaling (or shifting) pre-selected atmospheric vertical profiles. The vertical columns of the desired gases are determined from the measured sun-normalised radiance by

fitting a linearised radiative transfer model to it. A concise mathematical algorithm description and the key settings and adjustments for the simultaneous CO and CH$_4$ retrieval from TROPOMI's radiance measurements are summarised in the following subsections. The data products are based on TROPOMI Level 1b V01.00.00 files comprising spectra from the nominal operational mode, which started end of April, and reprocessed spectra from the previous six-month commissioning phase. The corresponding version is referred to as TROPOMI/WFMD (or WFMD in abbreviated form) v1.2.

## 2.1   Forward model

The forward model is derived from the radiative transfer model SCIATRAN (Rozanov et al., 2002, 2014) in pseudo-spherical atmosphere mode. To enable a fast retrieval, a look-up table scheme for the radiances and their derivatives has been implemented containing 17280 reference spectra for varying solar zenith angle, altitude, albedo, water vapour, and temperature. The reference spectra are computed with high spectral resolution in line-by-line mode and subsequently convolved to TROPOMI spec-

tral resolution of the SWIR bands using an instrument specific fixed spectral response function extracted from the TROPOMI ISRF Calibration Key Data v1.0.0 for nadir at $2338\,\mathrm{nm}$. The auxiliary input data include U.S. Standard atmosphere profiles




with methane scaled to $1850\,\mathrm{ppb}$, the SCIATRAN aerosol model using the background scenario described in Schneising et al. (2008, 2009), and HITRAN 2016 spectroscopic parameters (Gordon et al., 2017).

## 2.2 Inversion procedure

The linearised radiative transfer model (appropriately chosen from the look-up table according to the relevant parameters) plus a low order polynomial is linear least squares fitted to the logarithm of the measured sun-normalised radiance. The trace gas vertical profiles ($CH_4$, $CO$, $H_2O$) are scaled for the fit (i.e., the profile shape is not varied). Additional fit parameters are the shift of a pre-selected temperature profile, a scaling factor for the pressure profile, and parameters for a second order polynomial.

Let $m \in \mathbb{N}$ be the number of spectral points in the fitting window and $n \in \mathbb{N}$ the number of state vector elements (fit parameters) with $m \gg n$. The modelled radiance at wavelength $\lambda$ is given by

$$\ln I_\lambda^{mod}(\boldsymbol{v}, \boldsymbol{a}) = \ln I_\lambda^{mod}(\bar{\boldsymbol{v}}) + \sum_{j=1}^{n} \left.\frac{\partial \ln I_\lambda^{mod}}{\partial v_j}\right|_{\bar{v}_j} (v_j - \bar{v}_j) + P_\lambda(\boldsymbol{a}) \tag{1}$$

with state vector $\boldsymbol{v}$, linearisation point $\bar{\boldsymbol{v}}$, and polynomial coefficients $\boldsymbol{a}$ of second order polynomial $P$. A derivative with respect to a vertical column refers thereby to the change of the top-of-atmosphere radiance caused by a scaling of a pre-selected absorber concentration vertical profile. There are $m$ equations of this type, one for each detector pixel in the fitting window. The objective is to find the optimal state, so that the linear model best fits the observed radiance. This problem can be rewritten as

$$\boldsymbol{y} = \mathbf{A}\boldsymbol{x} + \boldsymbol{\epsilon} \tag{2}$$

with (log-)radiance difference $\boldsymbol{y} \in \mathbb{R}^m$ of measurement and linearised model due to a deviation $\boldsymbol{x} \in \mathbb{R}^n$ of the state vector from the multidimensional linearisation point, weighting function (Jacobian) matrix $\mathbf{A} \in \mathbb{R}^{m \times n}$ (with derivatives at the linearisation point and polynomial basis functions as columns), as well as sum of forward model error and (normally distributed log-transformed) instrument noise $\boldsymbol{\epsilon} \in \mathbb{R}^m$.

The covariance matrix associated with measurement noise is given by $\mathbf{C}_{\boldsymbol{y}} = \mathrm{diag}(\sigma_1^2, \ldots, \sigma_m^2) \in \mathbb{R}^{m \times m}$. To give larger weight to spectral points with smaller error variances and to obtain error estimates of the retrieval parameters via error propagation from the uncorrelated measurement errors $\sigma_i$, a weighted least squares approach is applied with matrix of weights defined by $\mathbf{W} = \mathbf{C}_{\boldsymbol{y}}^{-1}$. With the posterior probability $p(\boldsymbol{x}|\boldsymbol{y})$ of $\boldsymbol{x}$ given $\boldsymbol{y}$, the most probable inference of the inversion $\hat{\boldsymbol{x}} = \arg\max_{\boldsymbol{x} \in \mathbb{R}^n} p(\boldsymbol{x}|\boldsymbol{y})$ is obtained by minimising

$$f(\boldsymbol{x}) = \left\|\mathbf{W}^{\frac{1}{2}}(\boldsymbol{y} - \mathbf{A}\boldsymbol{x})\right\|_2^2 = (\boldsymbol{y} - \mathbf{A}\boldsymbol{x})^\mathrm{T}\mathbf{W}(\boldsymbol{y} - \mathbf{A}\boldsymbol{x}) \tag{3}$$

with respect to $\boldsymbol{x}$. Hence,

$$\frac{\partial f(\boldsymbol{x})}{\partial \boldsymbol{x}} = 2\left(\mathbf{A}^\mathrm{T}\mathbf{W}\mathbf{A}\boldsymbol{x} - \mathbf{A}^\mathrm{T}\mathbf{W}\boldsymbol{y}\right) \stackrel{!}{=} 0 \tag{4}$$





provides the solution $\hat{\boldsymbol{x}} = \mathbf{C}_{\boldsymbol{x}} \mathbf{A}^{\mathrm{T}} \mathbf{W} \boldsymbol{y}$ of the inverse problem, where $\mathbf{C}_{\boldsymbol{x}} = \left(\mathbf{A}^{\mathrm{T}} \mathbf{W} \mathbf{A}\right)^{-1}$ is the covariance matrix of solution $\hat{\boldsymbol{x}}$. The errors of the retrieval parameters are estimated by

$$\hat{\sigma}_j = \sqrt{(\mathbf{C}_{\boldsymbol{x}})_{jj}} \tag{5}$$

Due to the potential non-linear dependencies of the radiances with respect to water vapour and temperature within their
natural variability, the algorithm treats both parameters iteratively. The algorithm starts with look-up table elements representing U.S. Standard Atmosphere water vapour amount and temperature. If the retrieved parameter pair after the fit is closer to another look-up table element, the process is repeated with the corresponding reference spectrum. Usually convergence is achieved after one iteration step.

As the look-up table is only covering direct nadir conditions, a geometric path length correction has been implemented to
remove the path extension and associated enhancement of the retrieved vertical columns for off-nadir conditions with non-vanishing viewing zenith angle.

The spectral fitting windows in TROPOMI band 7 were optimised to retrieve CO and $CH_4$ simultaneously as accurately as possible (determined by an error analysis based on simulated measurements). They are shown in Figure 1 together with the absorption features of the relevant trace gases. Note that CO is a much weaker absorber compared to $CH_4$ and $H_2O$.
The apparent albedo is retrieved in the pre-processing by comparison of the measured continuum radiance with pre-calculated values from a look-up table. Cloud information is obtained from strong $H_2O$ absorption lines in band 8 (see Figure 1) by comparing the measured radiances to reference radiances for cloud-free conditions. As the absorption in these lines is strong, the measured radiance is small in the clear sky case. In the presence of clouds, most of the atmospheric $H_2O$ is shielded and the measured backscattered radiance coherently increases (Heymann et al., 2012). The corresponding ratio $r_{cld}$ of measured to
reference radiance for the selected strong absorption lines is thus an indicator of cloud contamination.

## 2.3 Sensitivity and error analysis using synthetic data

The sensitivity of the retrievals to different atmospheric layers is demonstrated by the vertical column averaging kernels (Figure 2). Compared to measurements in the thermal infrared spectral region, which are primarily sensitive to mid- or upper-tropospheric gas abundances in the absence of high thermal contrast, the advantage of the shortwave infrared spectral region is
the sensitivity to all altitude levels, including the boundary layer, which is important to analyse emissions originating from the Earth's surface.

As described in the previous subsection, the retrieval noise is determined via error propagation from the measurement noise. To assess the theoretical precision performance, we assume a simple shot noise limited noise model, which is defined in the following way: The reference signal-to-noise ratio is $SN_{ref} = 100$ in the continuum (radiance $L_{ref} = 4.3 \cdot 10^{11}$ phot/s/cm$^2$/nm/sr)
for a dark scene (albedo $= 0.05$) with low sun (solar zenith angle of $70°$) and is scaled according to

$$SN(L) = SN_{ref} \sqrt{\frac{L}{L_{ref}}} \tag{6}$$

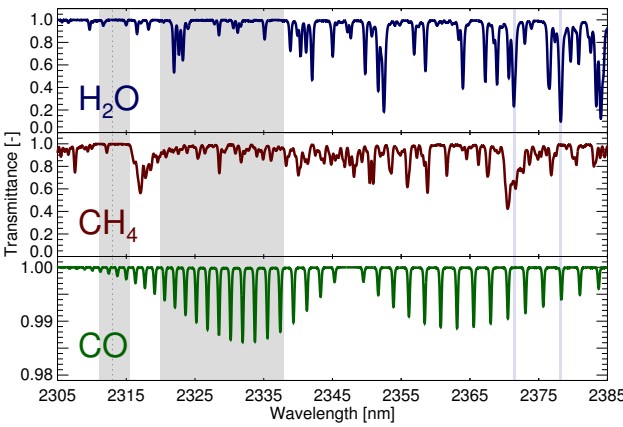

**Figure 1.** Fitting windows (grey) and trace gas transmittances for the SWIR bands of TROPOMI for U.S. Standard atmosphere concentrations. The strong $H_2O$ absorption lines between 2370 and 2380 nm used to obtain cloud information are shown in light blue . The apparent albedo is retrieved in the continuum at 2313 nm (dashed line).

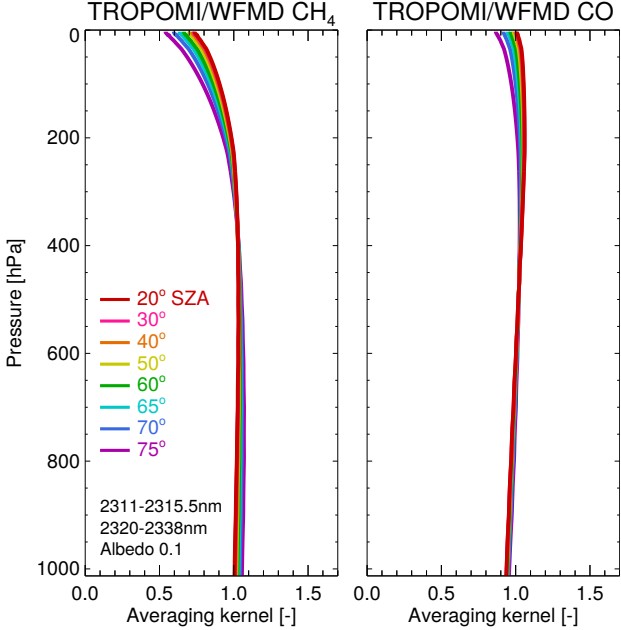

**Figure 2.** $CH_4$ and CO column averaging kernels reflecting the altitude sensitivity of the retrievals.

for other radiances. The resulting absolute precision is widely independent of the current concentrations. For U.S. Standard atmosphere values, the corresponding relative retrieval noise for different albedos and solar zenith angles is shown in Figure 3. It is below 1% for solar zenith angles smaller than 75° and albedos larger than 0.03 in the case of $CH_4$. As the CO absorption





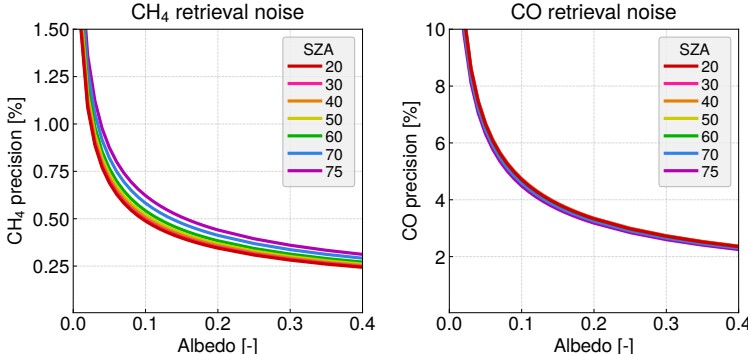

**Figure 3.** TROPOMI/WFM-DOAS $CH_4$ and CO relative retrieval noise for U.S. Standard atmosphere conditions.

is considerably weaker than the $CH_4$ absorption, the CO retrieval exhibits larger relative noise, which is below $8\%$ for albedos larger than $0.03$.

The analysis of systematic errors is performed using simulated measurements. That means that for different scenarios defined by specific atmospheric conditions, radiances and irradiances are calculated with the radiative transfer model, which are
subsequently used as measurement input in the retrieval. The errors are then defined as the deviation of the retrieved from the true quantities. The corresponding results for several scenarios are summarised in Table 2. All scenarios already include interpolation between different wavelength grids (for measured and reference spectra) unless otherwise stated.

The analysis includes *Basic* scenarios testing if perturbations of the state vector elements can be retrieved, quantifying look-up table interpolation errors, and analysing errors caused by off-nadir conditions. In order to examine the sensitivity to
vertical profile variations, the scenario class of *Profiles* includes several model atmospheres (extracted from MODTRAN (Berk et al., 1998)), which differ from the U.S. Standard Atmosphere with respect to temperature, pressure, water vapour, carbon monoxide, and methane profiles. These scenarios are more difficult to deal with than the basic ones, because the perturbations are not consistent with the scaling assumption, i.e., they include proper variations of the profile shape.

Also examined is the sensitivity to the *Spectral albedo* of the natural surface types shown in Figure 4 taken from the ASTER
and USGS spectral libraries. The analysed *Aerosol* scenarios are largely described in Schneising et al. (2008, 2009) with aerosol type definitions in the different atmospheric layers based on Optical Properties of Aerosols and Clouds (OPAC) (Hess et al., 1998). The retrieval errors due to undetected *Subvisual clouds* are also investigated for different ice and water clouds.

This gives an impression of the magnitude of errors one can expect assuming that thick clouds can be filtered out by cloud screening in the pre- or post-processing: Typical systematic retrieval errors are below $1\%$ for methane and below $2\%$ for carbon
monoxide even for challenging scenarios.

Larger systematic errors in the case of thick clouds are expected because clouds are not explicitly considered in the forward model of the retrieval algorithm to retain the high processing speed. Therefore, the systematic biases due to clouds are further analysed in more detail. The results for water and ice clouds at different heights are summarised in Figure 5. Thereby, clouds



**Table 2.** Error analysis for different scenarios. Standard settings are direct nadir, sea level, solar zenith angle $50°$, albedo 0.1, and U.S. Standard atmosphere. Scenarios with $\oplus$ include scaling of the $CH_4$ and CO profiles by 10%, for scenarios with $\angle$ the sensor zenith angle is set to $30°$ (relative azimuth $60°$). Standard cirrus are located between 11 and $12\,km$ ($\tau=0.03$) consisting of fractal ice crystals with an edge length of $100\,\mu m$. Standard cumulus are located between 3 and $4\,km$ ($\tau=0.03$) consisting of water droplets with an effective radius of $10\,\mu m$.

| | scenario | $CH_4$ error [%] | CO error [%] |
|---|---|---|---|
| Basic | dry run (no $\lambda$ interpol.) | 0.00 | 0.00 |
| | dry run | 0.00 | −0.03 |
| | dry run $\oplus$ | −0.08 | −0.15 |
| | dry run $\angle$ | −0.09 | −0.20 |
| | $T + 30\,K$ | 0.25 | −0.24 |
| | $T - 30\,K$ | 0.06 | −0.42 |
| | $p + 5\%$ | −0.01 | −0.06 |
| | $p - 5\%$ | −0.04 | −0.10 |
| | albedo 0.2 | −0.01 | −0.04 |
| Profiles | midlatitude summer | 0.12 | 0.35 |
| | midlatitude winter | −0.13 | 0.68 |
| | subarctic summer | 0.09 | 0.60 |
| | subarctic winter | 0.63 | −0.59 |
| | tropical | 0.15 | −0.94 |
| Spectral albedo | sand | −0.03 | −0.04 |
| | soil | 0.01 | −0.03 |
| | rangeland | 0.02 | −0.11 |
| | deciduous | −0.07 | 0.01 |
| | conifers | 0.01 | −0.19 |
| | snow | −0.25 | −0.30 |
| | ocean | 0.00 | −0.07 |
| Aerosols | no aerosol | 0.01 | 0.10 |
| | urban | 0.11 | 0.04 |
| | desert (sand albedo) | 0.41 | 0.40 |
| | arctic (snow albedo) | −0.19 | −0.41 |
| | extreme in BL | 0.34 | −0.43 |
| | extreme in BL $\oplus$ | 0.24 | −0.51 |
| | extreme in BL $\angle$ | −0.28 | −1.38 |
| Subvisual clouds | cirrus | −0.29 | −0.87 |
| | cirrus $\oplus$ | −0.41 | −0.99 |
| | cirrus $\angle$ | −0.86 | −1.84 |
| | cirrus (fractal 50) | −0.33 | −0.94 |
| | cirrus (hexagonal $50 \times 100$) | 0.11 | −0.20 |
| | cirrus ($\tau=0.05$) | −0.56 | −1.53 |
| | cumulus | −0.31 | −0.86 |
| | cumulus ($R=6\,\mu m$) | −0.21 | −0.74 |
| | cumulus ($R=14\,\mu m$) | −0.32 | −0.87 |
| | cumulus ($\tau=0.05$) | −0.55 | −1.44 |





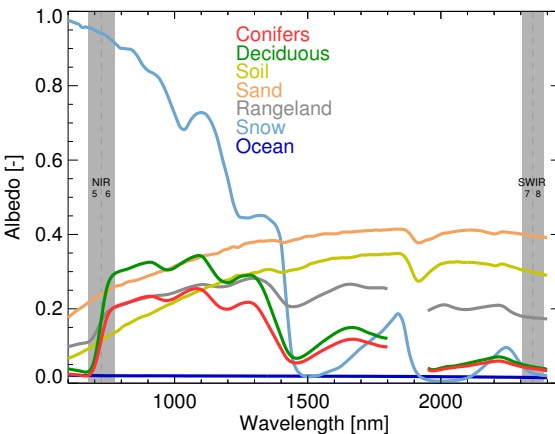

**Figure 4.** Spectral albedos of different natural surface types. Reproduced from the ASTER Spectral Library through the courtesy of the Jet Propulsion Laboratory, California Institute of Technology, Pasadena, California (©1999, California Institute of Technology) and the Digital Spectral Library 06 of the U.S. Geological Survey.

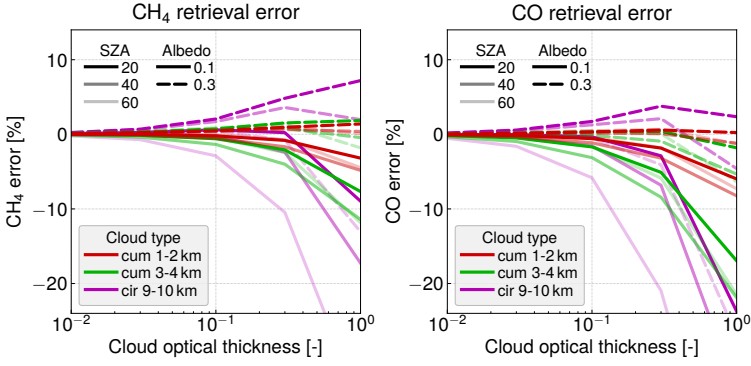

**Figure 5.** Systematic retrieval errors for different water and ice clouds under various conditions.

are modelled as a layer of $1\,km$ vertical extent consisting of water droplets with an effective radius of $10\,\mu m$ or fractal ice crystals with an edge length of $100\,\mu m$. The analysis is performed for three different clouds types: two water clouds with cloud top heights (CTH) of 2 and $4\,km$ and an ice cloud with CTH of $10\,km$.

As expected, the absolute value of systematic errors typically increases with increasing cloud optical thickness, increasing
5   cloud top height, increasing solar zenith angle, and decreasing albedo. In most cases, there is a considerable underestimation of the vertical column in the case of thick clouds. However, there are also conditions, where the absolute value of the error is small even at a cloud optical thickness of $\tau{=}1$ or occasionally turns to an overestimation for measurements over bright surfaces. Overall the systematic errors due to clouds are qualitatively similar for CO and $CH_4$.





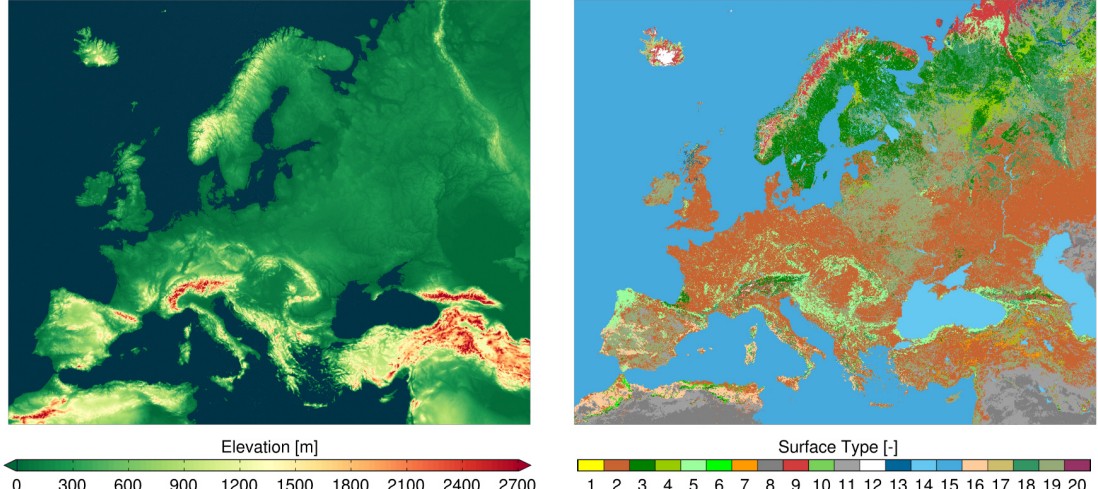

**Figure 6.** USGS GMTED2010 elevation and GLCC surface type (Biosphere Atmosphere Transfer Scheme Legend) with $0.05°$ resolution using the example of Europe. In the GMTED2010 data set, ocean areas have been assigned a value of 0 (shown in dark blue).

The error analysis based on synthetic data has shown that perturbations of the state vector elements can be clearly retrieved and that the algorithm is theoretically suitable to successfully retrieve carbon monoxide and methane from real TROPOMI data for cloud-free scenes. In the case of thick clouds, systematic errors can become rather large confirming that an efficient cloud screening algorithm is necessary, in particular to meet the demanding requirements on the precision and accuracy of atmo-
spheric $CH_4$ measurements. An appropriate quality filter is implemented in the post-processing and described in Section 2.5.2. For CO it may be possible to relax the filter due to the less stringent requirements, but for now we employ a joint quality filter for both simultaneously retrieved trace gases.

## 2.4   High resolution auxiliary data

As a consequence of the high spatial resolution of the TROPOMI SWIR measurements, a digital elevation model required
for selection and interpolation of suitable pre-calculated reference spectra and a land cover characterisation data set necessary to provide land fraction and surface type as additional information have also to be implemented in high resolution. For this purpose, the Global Multi-resolution Terrain Elevation Data 2010 (GMTED2010) and the Global Land Cover Characterization (GLCC) of the United States Geological Survey (USGS) (United States Geological Survey, 2018a, b) are used with a resampled resolution of $0.05°$ (about $5\,\mathrm{km}$ at the equator) to compute surface elevation, land fraction, and dominating surface type
(Biosphere Atmosphere Transfer Scheme Legend) for every sounding of the satellite.

Some incorrect values of zero elevation in the GMTED2010 data set over the Caspian Sea and Lake Superior have been replaced with corresponding Global 30 Arc-Second Elevation (GTOPO30) values (United States Geological Survey, 2018c). Figure 6 demonstrates the resolution of the implemented elevation and surface type data sets using the example of Europe.





## 2.5 Post-processing

### 2.5.1 Column-averaged dry air mole fractions

In order to convert the retrieved vertical columns into column-averaged dry air mole fractions (denoted XCO and $XCH_4$), the columns are divided by the dry air column obtained from the European Centre for Medium-Range Weather Forecasts

(ECMWF) analysis. Thereby, the ECMWF dry columns are corrected for the actual surface elevation of the individual TROPOMI measurements (based on the deviation from the mean altitude of the coarser model grid) inheriting the high spatial resolution of the satellite data.

An analysis based on simulated measurements has indicated that this approach is superior to a normalisation by simultaneously retrieved oxygen ($O_2$ A-band) from TROPOMI band 6 for off-nadir conditions and/or in the presence of strong scatterers

in the atmosphere (aerosol, clouds) as a consequence of the spectral distance in combination with the albedo differences of natural surface types between NIR band 6 and SWIR band 7 (see Figure 4). For these reasons, $O_2$ is a barely sufficient proxy for the lightpath in the $2.3\,\mu m$ spectral range in a scattering atmosphere. In addition to the better accuracy of the ECMWF-based mole fraction computation, this approach is also faster, because the oxygen fit and the interband coregistration mapping can be omitted. The out-of-spectral-band straylight issue of the TROPOMI band 6 (Kleipool et al., 2018) would potentially further

hamper the $O_2$-proxy approach.

### 2.5.2 Quality filter

To enable a fast processing speed to handle the huge amount of TROPOMI data, the look-up table is limited to rather simple physical conditions (e.g., cloud-free scenes). Thus, a quality screening algorithm excluding measurements not sufficiently characterized by the forward model had to be implemented. First of all, challenging conditions with solar zenith angles larger

than $75°$, which are increasingly prone to scattering and saturation related issues due to the weakening signal and lengthening of the light path, are cut off. To be independent of other data sets and their ongoing availability, it was aimed at filtering based on parameters directly included in the retrieval output. This was achieved by using a machine learning approach based on a random forest classifier, which is a meta estimator growing many independent decision trees on different subsamples of the data set and uses averaging to improve the predictive accuracy and prevent overfitting. Thereby, each tree of the ensemble is

grown in the following way (Breiman, 2001):

1. Randomly draw $N$ samples from the training set of size $N$ with replacement (bootstrap sample). For large $N$, a fraction of about $63.2\%$ unique samples is expected, the remainder being duplicates.

2. From the $F$ input variables, $f \ll F$ are randomly chosen out of $F$ and the best split according to minimisation of Gini impurity on these $f$ is used to split the node (Breiman, 1996a). The value of $f$ is held constant during the forest growing.

3. There is no pruning of the decision trees, i.e., each tree is grown to the largest possible extent.



To classify a new previously unseen measurement after growing the forest with the training data, each decision tree gives a classification according to the input features of the measurement and the forest chooses the majority vote over all trees in the forest. The combination of the tree results each based on different bootstrap replicates of the learning set is called *bootstrap aggregating* or *bagging* (Breiman, 1996b). The forest error rate depends on the *correlation* between the trees in the forest and

the *strength* of the individual trees. The forest error rate decreases with decreasing correlation and increasing strength of the trees. Reducing $f$ reduces both the correlation and the strength, while increasing $f$ increases both. Hence, there is an optimal range of $f$ minimising the forest error rate.

We use a forest size of 200 trees and the well-recognised standard choice of $f = \sqrt{F}$. The training data set comprises 16 randomly chosen days, namely 25 November in 2017, as well as 25 February, 21 April, 27 April, 22 May, 28 May, 16 June,

26 June, 16 July, 24 July, 3 August, 22 August, 18 September, 4 October, 21 October, and 16 November in 2018. For each day, 5 million measurements are randomly selected. Thus, the training subset consists of 80 million measurements, which are classified based on cloud information from VIIRS onboard Suomi NPP (Hutchison and Cracknell, 2005), which flies in loose formation configuration with Sentinel-5 Precursor (S5P trails behind by 3.5 minutes). This classification is augmented by additionally flagging distinct $XCH_4$ deviations relative to a climatology consisting of MACC-2 flux inversion system (Berga-

maschi et al., 2013) averages on a $6° \times 4°$ grid for the years 2003-2005 adjusted by an accumulated increase until the time of the measurement based on globally averaged marine NOAA surface data (Dlugokencky, 2018), identifying scenes obviously not well characterized by the forward model, in particular conspicuously decreased methane abundances in the presence of clouds due to shielding of the underlying atmosphere or in the case of very low surface reflectances.

To train the forest, a set $F$ of 25 feature variables is selected by feature ranking with recursive feature elimination and

cross-validated selection of the best features. $20\%$ of the training data are randomly drawn and retained as test data. The corresponding predictive accuracy is shown in Figure 7 as function of the selected features confirming that the random forest does not overfit, because the accuracy has its global maximum when using all 25 features. The selected variables in order of importance are: 1.) $H_2O$ column difference to ECMWF, 2.) cloud parameter $r_{cld}$, 3.) simplified surface type (water, coastal, land, desert, ice), 4.) linear polynomial coefficient $p_1$, 5.) pressure difference to ECMWF, , 6.) altitude, 7.) latitude, 8.) CO fit

error, 9.) temperature, 10.) root mean square of fit residual, 11.) temperature difference to ECMWF, 12.) $H_2O$ fit error, 13.) pressure fit error, 14.) $H_2O$ column, 15.) longitude, 16.) solar zenith angle, 17.) pressure, 18.) quadratic polynomial coefficient $p_2$, 19.) radiance ratio strong $H_2O$ absorption to continuum, 20.) dry air column from ECMWF, 21.) retrieved apparent albedo, 22.) continuum radiance, 23.) relative azimuth angle, 24.) across-track dimension index, and 25.) strong $H_2O$ absorption radiance. The predictive accuracy when using all 25 features amounts to $0.983$, which means that $98.3\%$ of all scenes are

correctly classified.

A more detailed analysis of the predictive power of the random forest can be obtained from the confusion matrix of the test data set (also shown in Figure 7). As can be seen, the data set is unbalanced with barely $10\%$ belonging to class 0 denoting the good measurements. This is primarily due to the large amount of cloudy scenes in combination with the issue that mainly sun glint or glitter scenes are classified as good over the ocean and inland waters as a consequence of the weak signal ascribed

to the low reflectances of these dark surfaces (see Figure 4). For land scenes the fraction of good observations accordingly



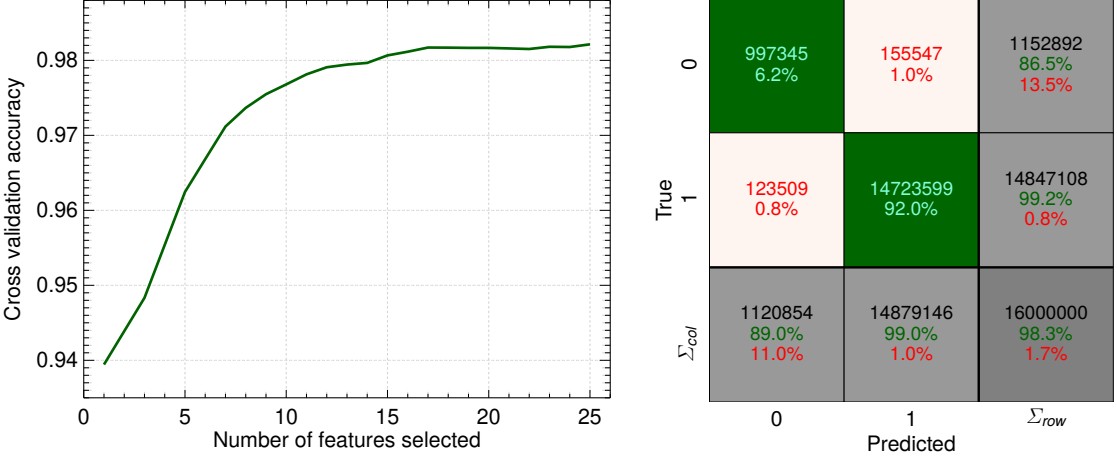

**Figure 7.** Cross validated predictive accuracy of the quality classification random forest obtained by recursive feature elimination. The confusion matrix when using all 25 features is shown on the right hand side for the test data set denoting good observations with 0 and measurements to be excluded with 1. The green diagonal cells correspond to correct classifications, the red off-diagonal cells to incorrect classifications. The number of scenes and the percentage of the total number of scenes are given in each cell. Important key parameters are summarised in the grey cells along the edge. The right column shows the percentages of all the elements belonging to each class that are correctly (recall) and incorrectly (false negative rate) classified, the bottom row those which are predicted to belong to each class that are correctly (precision) and incorrectly (false discovery rate) classified. The dark grey cell in the bottom right corner displays the overall accuracy.

increases to about 20%. The percentage of all good measurements that are incorrectly excluded (false negative rate of class 0) amounts to about 13% (9% for land scenes). For these cases the filter is too strict, but the quality of the data passing the filter is not compromised. The percentage of all the measurements predicted to be good that are incorrectly classified and should actually be excluded (false discovery rate of class 0) amounts to about 11%. For these cases the filter appears not stringent

5   enough. However, as the training classification is quite strict, that does not necessarily mean that all these measurements are actually of low quality. The rate can rather be interpreted as an upper bound of potentially remaining challenging retrievals on the verge of sufficient characterisation by the forward model, e.g., observations near cloud edges. The effective diagnostic performance of the quality filter will emerge from the validation.

Adding additional parameters to $F$ does not significantly improve the predictive accuracy further. It is important to note that

10   the resulting classification is independent of the absolute abundances of the primary retrieval parameters $CH_4$ and $CO$. The performance of the classification algorithm is exemplary demonstrated in Figure 8 confirming that cloudy scenes are reliably excluded in general and that the quality filter is usually stricter than the VIIRS classification, in particular over the weakly reflecting ocean. Measurements classified as cloudy by VIIRS but still passing the quality filter are rare and not associated with conspicuous methane abundances.

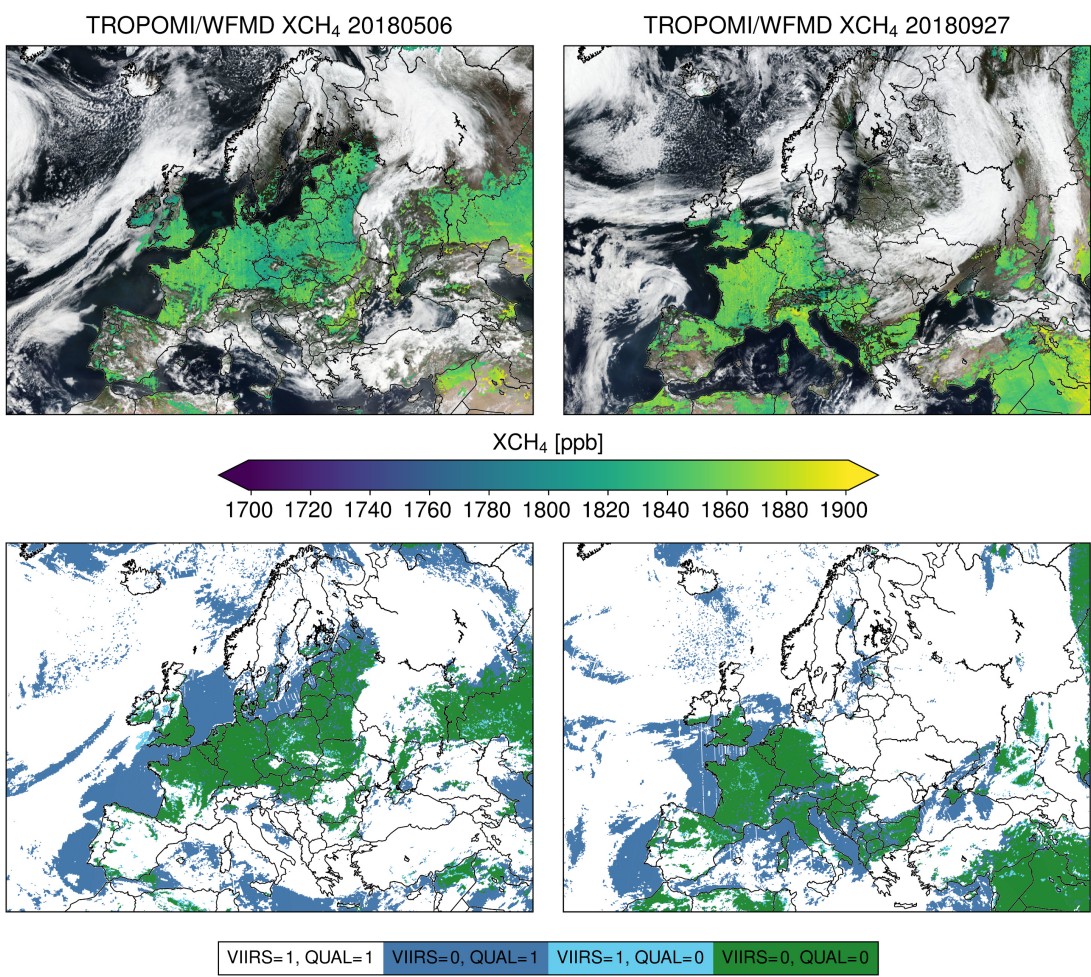

**Figure 8.** Top: Quality filtered XCH$_4$ over Europe overlayed on true color reflectances from the Visible Infrared Imaging Radiometer Suite (VIIRS) for two example days not included in the training data set demonstrating the performance of the machine learning classification algorithm. Evidently, cloudy scenes are typically identified and excluded. The VIIRS images are taken from the NASA Worldview application. Bottom: Comparison of the implemented quality filter with the VIIRS cloud classification (1=cloudy). Matching classifications are shown in white and green. By definition the quality filter is generally stricter than the VIIRS cloud flag and the blue areas are additionally excluded. The rare instances of measurements classified as cloudy by VIIRS but still passing the quality filter are shown in cyan.

### 2.5.3 Shallow learning calibration for methane

The implemented machine learning-based quality filter described in the previous subsection removes observations not sufficiently characterized by the forward model. Although this procedure typically excludes scenes exhibiting large systematic errors, smaller systematic errors may remain in the residual data set. In particular, there seems to be a systematic albedo dependence of unknown origin of retrieved methane abundances with an underestimation over dark surfaces. As a consequence





of the fairly stringent quality requirements for methane, a random forest regressor algorithm was implemented to reduce the remaining systematic methane errors after the retrieval by calibrating against an assumed standard defined below, which is deemed insensitive to surface reflectance variations.

Like the classification algorithm described in the previous subsection, the random forest regressor (Criminisi and Shotton,
2013) grows an ensemble of decision trees, training each tree on a different data sample applying the bootstrap aggregating technique. From the $f$ randomly chosen parameters the optimal split maximising the variance reduction in the child nodes is used to split the nodes. To focus on the most prominent features (shallow learning of systematic errors caused by surface albedo variations), the tree growing is limited to 500 leaf nodes. Again, a forest size of 200 trees and $f = \sqrt{F}$ is used, where $F$ consists of 5 feature variables, which are in order of importance: retrieved apparent albedo, solar zenith angle, cloud parameter
$r_{cld}$, strong $H_2O$ absorption radiance, and across-track dimension index.

To compute the correction for a new measurement after growing the forest with the training data, each decision tree provides a regression according to the input features of the measurement and the random forest uses the average over all tree regressions as final calibration value for this observation. In other words, the random forest regressor uses averaging in the bagging procedure to combine the individual tree results (in contrast to voting used in the classification case).
The calibration data set consists of the $XCH_4$ climatology introduced in the previous subsection evaluated for selected regions spanning a wide range of albedos and solar zenith angles. For individual regions, the climatology is roughly corrected for potential systematic overall biases by adding up a single region-specific correction value based on a comparison to nearby sites of the Total Carbon Column Observing Network (TCCON) (Wunch et al., 2011a) for the year 2017. In any case, the seasonal and intra-regional spatial variations are solely determined by the climatology. The training regions and corresponding
climatology correction values are shown in Figure 9.

The standard deviation of the resulting $XCH_4$ correction when considering global yearly averages of gridded data (on a $0.1° \times 0.1°$ grid) amounts to 13 ppb, which is well below the natural variability. The XCO data set is not corrected.

## 3   Validation

TCCON is a network of ground-based Fourier-transform spectrometers recording direct solar spectra in the NIR/SWIR spectral
region to retrieve accurate and precise column-averaged abundances of several atmospheric constituents, including XCO and $XCH_4$, thus providing a validation resource for satellite data (Wunch et al., 2011a). To ensure comparability, all TCCON sites use similar instrumentation (Bruker IFS 125HR) and a common retrieval algorithm. The TCCON data are tied to the WMO trace gas scale using airborne in situ measurements applying individual scaling factors for each species. The estimated accuracy ($1\sigma$) is about 2 ppb for XCO and 3.5 ppb for $XCH_4$ (Wunch et al., 2010).
To compare the satellite data with TCCON quantitatively, it has to be taken into account that the sensitivities of the instruments differ from each other and that individual apriori profiles are used to determine the best estimate of the true atmospheric state, respectively. The first step is to correct for the apriori contribution to the smoothing equation by adjusting the measurements for a common apriori profile (Rodgers, 2000; Schneising et al., 2012; Dils et al., 2014). Here we use the TCCON prior





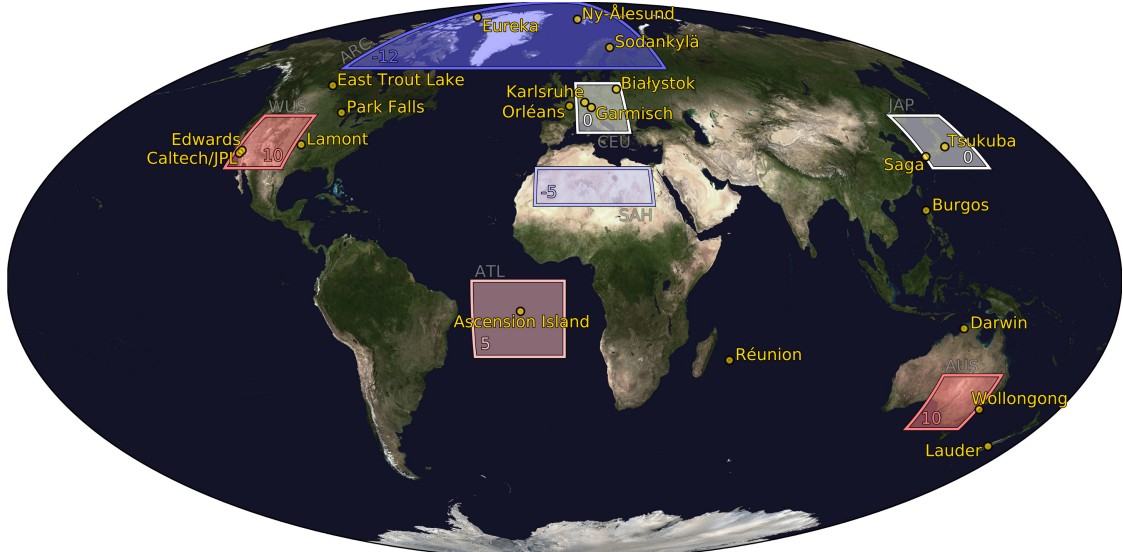

**Figure 9.** Regions used to train the machine learning regressor comprising Arctic (ARC), Western United States (WUS), Central Europe (CEU), Japan (JAP), Sahara (SAH), South Atlantic (ATL), and Australia (AUS). The corresponding numbers specify the regional corrections applied to the methane climatology before learning in ppb, which are also colour-coded in the borders and backgrounds of the regions (blue for negative and red for positive corrections). The yellow circles highlight the TCCON sites used in the validation.

as the common apriori profile for all measurements:

$$\hat{c}_{\mathrm{adj}} = \hat{c} + \frac{1}{m_0} \sum_l m_l \left(1 - A_l\right) \left(x_{a,T}^l - x_a^l\right) \tag{7}$$

In this equation, $\hat{c}$ represents the originally retrieved TROPOMI column-averaged dry air mole fraction, $l$ is the index of the vertical layer, $A_l$ the corresponding column averaging kernel of the TROPOMI algorithm, $\boldsymbol{x}_a$ and $\boldsymbol{x}_{a,T}$ the TROPOMI and TCCON apriori dry air mole fraction profiles. $m_l$ is the mass of dry air determined from the dry air pressure difference between the upper and lower boundary of layer $l$ via $\frac{\Delta p_l}{g_l}$ with gravitational acceleration $g_l$ and $m_0 = \sum_l m_l$ is the total mass of dry air. To minimise the smoothing error introduced by the averaging kernels we do not compare $\hat{c}_{\mathrm{adj}}$ directly with the retrieved TCCON mole fractions $\hat{c}_T$ but rather with the adjusted expression (Rodgers and Connor, 2003; Wunch et al., 2011b)

$$\hat{c}_{T,\mathrm{adj}} = c_{a,T} + \left(\frac{\hat{c}_T}{c_{a,T}} - 1\right) \frac{1}{m_0} \sum_l m_l A_l x_{a,T}^l \tag{8}$$

Thereby, $c_{a,T}$ represents the TCCON apriori column-averaged dry air mole fraction associated with the apriori profile $\boldsymbol{x}_{a,T}$. However, using $\hat{c}_{T,\mathrm{adj}}$ instead of $\hat{c}_T$ has only marginal impact on the validation results presented here, because the satellite averaging kernels are close to 1 in the lower atmosphere (see Figure 2) implying $\hat{c}_{T,\mathrm{adj}} \approx \hat{c}_T$.

The validation is performed at the TCCON sites listed in Table 3 (see also Figure 9). For the comparison a set of collocation criteria has to be specified. Ideally, the representativity is maximised by as strict as possible criteria while concurrently ensuring



**Table 3.** TCCON sites used in the validation ordered according to latitude from north to south.

| Station | Latitude [°] | Longitude [°] | Altitude [km] | Reference |
|---|---|---|---|---|
| Eureka | 80.05 | −86.42 | 0.61 | Strong et al. (2019) |
| Ny-Ålesund | 78.92 | 11.92 | 0.02 | Notholt et al. (2017) |
| Sodankylä | 67.37 | 26.63 | 0.19 | Kivi et al. (2014); Kivi and Heikkinen (2016) |
| East Trout Lake | 54.35 | −104.99 | 0.50 | Wunch et al. (2018) |
| Białystok | 53.23 | 23.03 | 0.19 | Deutscher et al. (2015); Messerschmidt et al. (2012) |
| Karlsruhe | 49.10 | 8.44 | 0.11 | Hase et al. (2015) |
| Orléans | 47.97 | 2.11 | 0.13 | Warneke et al. (2014) |
| Garmisch | 47.48 | 11.06 | 0.75 | Sussmann and Rettinger (2018a) |
| Park Falls | 45.94 | −90.27 | 0.44 | Wennberg et al. (2017); Washenfelder et al. (2006) |
| Lamont | 36.60 | −97.49 | 0.32 | Wennberg et al. (2016b) |
| Tsukuba | 36.05 | 140.12 | 0.03 | Morino et al. (2018a) |
| Edwards | 34.96 | −117.88 | 0.70 | Iraci et al. (2016) |
| JPL | 34.20 | −118.18 | 0.39 | Wennberg et al. (2016a) |
| Caltech | 34.14 | −118.13 | 0.24 | Wennberg et al. (2015) |
| Saga | 33.24 | 130.29 | 0.01 | Shiomi et al. (2014) |
| Burgos | 18.53 | 120.65 | 0.04 | Morino et al. (2018b); Velazco et al. (2017) |
| Ascension Island | −7.92 | −14.33 | 0.03 | Feist et al. (2014) |
| Darwin | −12.46 | 130.93 | 0.04 | Griffith et al. (2014a); Deutscher et al. (2010) |
| Réunion | −20.90 | 55.49 | 0.09 | De Mazière et al. (2017) |
| Wollongong | −34.41 | 150.88 | 0.03 | Griffith et al. (2014b) |
| Lauder | −45.04 | 169.68 | 0.37 | Sherlock et al. (2014); Pollard et al. (2017) |

sufficient data for a sound and stable comparison. This trade-off is resolved by the following selection. The spatial collocation criterion requires the satellite measurements to lie within a radius of $100\,\mathrm{km}$ around the TCCON site and that the altitude difference is smaller than $250\,\mathrm{m}$. The temporal collocation criterion is set to $\pm2$ hours. As a consequence of the altitude representativity criterion, there are not enough collocations for a robust comparison at the mountain sites Zugspitze (Sussmann

5    and Rettinger, 2018b) and Izaña (Blumenstock et al., 2017).

The validation results are summarised in Figures 10 and 11 including the mean bias $\mu$ and the scatter $\sigma$ relative to TCCON for each site. The parameter $\sigma$ is estimated from Huber's Proposal-2 M-estimator (Huber, 1981), which is a well established estimator of location and scale being robust against outliers of a normal distribution. This is an appropriate choice and preferred over the standard deviation, because one is interested in the actual single measurement precision without distortion of the results

10   by a few outliers, which are rather attributed to systematic errors, e.g. due to residual clouds. As a consequence, outliers are





fully included in the computation of the systematic error but get lower weight in the robust determination of the random error, which is interpreted as a measure of the repeatability of measurements.

It is also checked whether the respective site biases are sensitive to the selection of the spatial collocation radius, which is an indication of sources within the satellite collocation area with only marginal influence on the TCCON measurements itself. A considerable sensitivity was found for $XCH_4$ at Edwards. The collocation region intersects oil production areas in California's Central Valley (in contrast to Caltech and JPL, see also results in Section 4 and Figure 23) as well as the South Coast Air Basin (SoCAB), which has a well-known methane enhancement (Wunch et al., 2016). As such nearby sources limit the representativity of affected satellite measurements, the collocation radius is reduced to 50 km for Edwards.

The altitude representativity criterion separates the well-isolated air masses of the SoCAB, where Caltech and JPL are located, from the Mojave desert with the Edwards site to the north. Hence, different air masses are analysed in the validation at Caltech/JPL and Edwards, although the corresponding collocation circles overlap. This also explains the insensitivity to the spatial collocation radius at Caltech/JPL and why no additional constraints on the coincidence criteria are necessary for these sites to ensure representativity. As Caltech and JPL are both exposed to SoCAB air masses, the permissible altitude collocation tolerance of Caltech is equally assumed for JPL despite slightly differing surface elevation.

The results for the individual sites are condensed to the following parameters for the overall quality assessment of the satellite data: the global offset is defined as the mean of the local offsets at the individual sites, the random error is the global scatter (analogously estimated to the single site case) of the differences to TCCON after subtraction of the respective regional biases, and the systematic error is the standard deviation of the local offsets relative to TCCON at the individual sites as a measure of the station-to-station biases. For XCO the global offset amounts to 4.46 ppb, the random error is estimated to be 5.13 ppb (6.12 ppb when using the standard deviation instead of Huber's Proposal-2 M-estimator), and the systematic error is 1.88 ppb, which is on the order of the estimated (station-to-station) accuracy of the TCCON of about 2 ppb. For $XCH_4$ the global offset aggregates to $-1.39$ ppb, the random error is 14.02 ppb (15.79 ppb when using the standard deviation), and the systematic error is given by 4.36 ppb, which is again similar to the TCCON accuracy of about 3.5 ppb.

To further analyse how well the real temporal and spatial variations are captured by the TROPOMI data, Figure 12 shows a comparison to TCCON based on daily means for days with more than three collocations. The obvious linear relationship with a high correlation for both gases ($R = 0.97$ for XCO and $R = 0.91$ for $XCH_4$) underlines the typical good agreement of the satellite and validation data. The linear regression yields a fit close to the 1:1 line for both gases.

In the case of $XCH_4$, there are a few outliers where the satellite values are considerably lower than the TCCON values. These occasional instances are not site specific and can probably be ascribed to days with residual or partial cloud cover interfering with the satellite retrievals. Outliers with higher values compared to TCCON are more rare and dominated by a handful of collocations at East Trout Lake. This exceptional lack of $XCH_4$ agreement occurs on four days in the time period February 10-21 as well as on March 29 and may be attributable to Arctic polar vortex air above East Trout Lake potentially causing the following related issues: associated fronts of different air masses may complicate the identification of collocations near the vortex edge and/or the stratospheric part of the methane profile may be largely affected by the polar vortex leading to a considerable deviation from the assumed apriori profile shapes (Tukiainen et al., 2016). It is verified that the impact of outliers

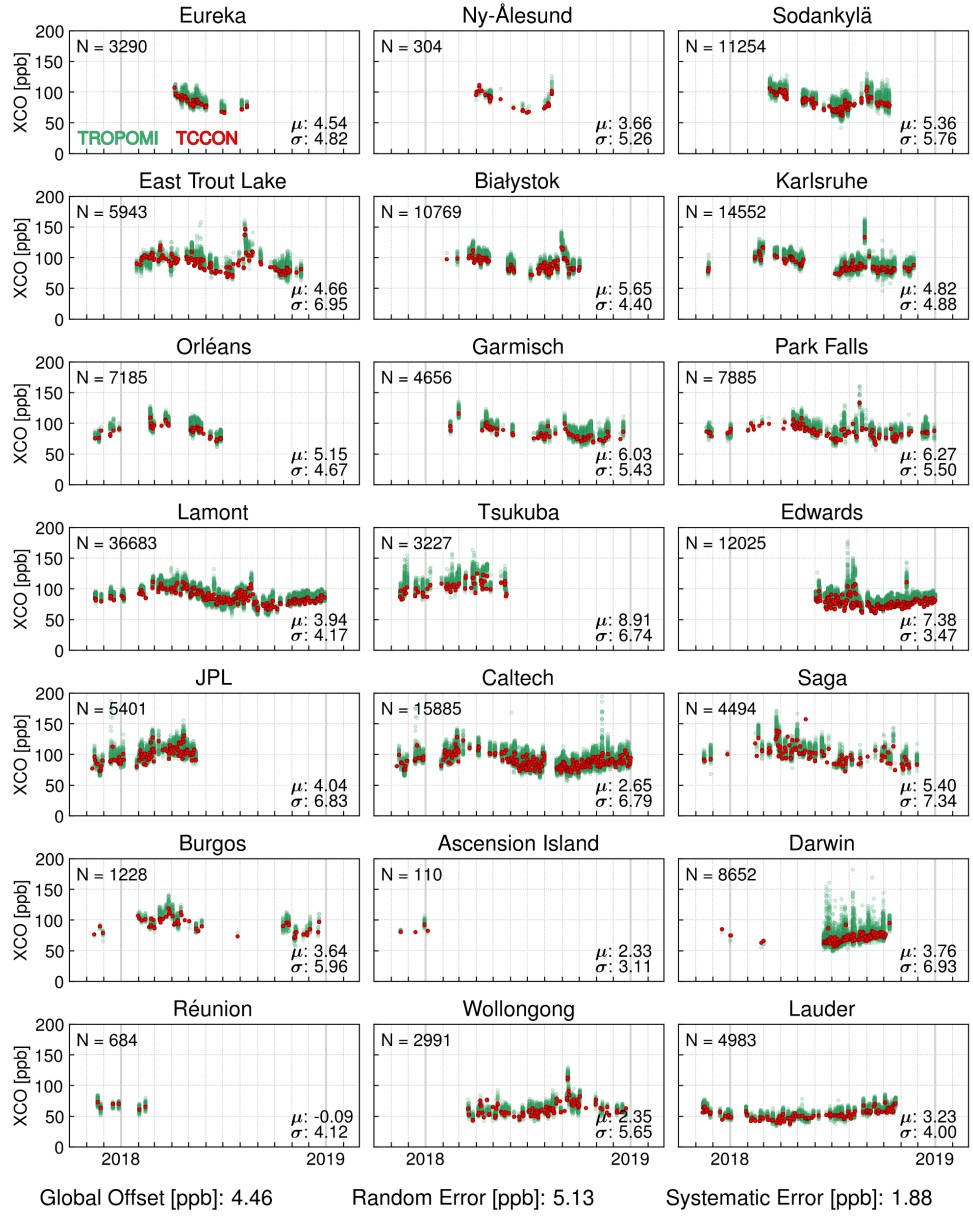

**Figure 10.** Comparison of the TROPOMI/WFMD v1.2 XCO time series (green) with ground based measurements from the TCCON (red). For each site, $N$ is the number of collocations, $\mu$ corresponds to the mean bias and $\sigma$ to the scatter of the satellite data relative to TCCON in ppb. $\sigma$ is estimated from Huber's Proposal-2 M-estimator. The global offset is defined as the mean of the local offsets at the individual sites, the random error is the global scatter of the differences to TCCON after subtraction of the respective regional biases, and the systematic error is the standard deviation of the $\mu$ at the individual sites.



**Figure 11.** As Figure 10 but for $XCH_4$.

on the regression is marginal by repeating the fit with the Huber linear regression model (Huber and Ronchetti, 2009), which

is robust to outliers and provides similar results to the standard linear regression here.





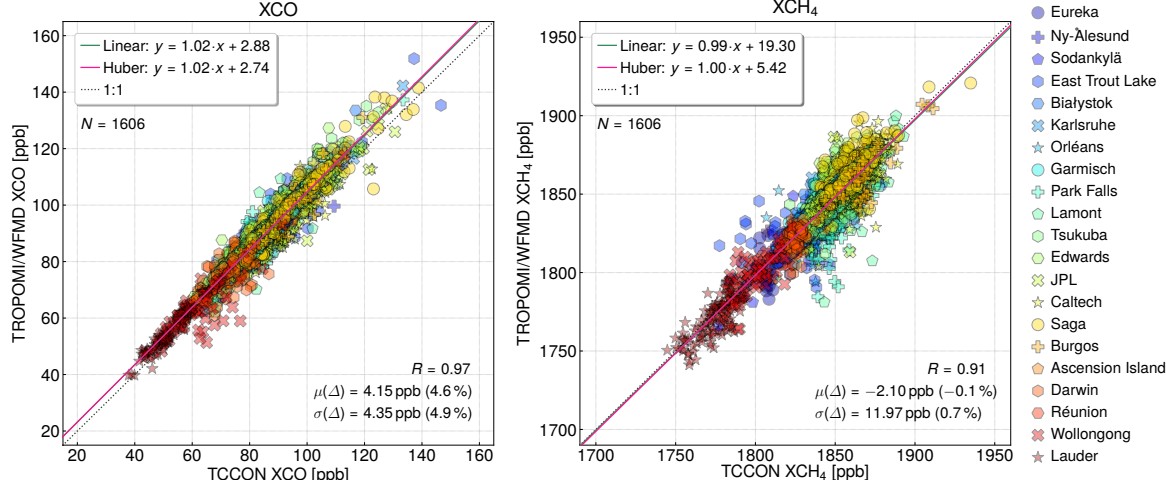

**Figure 12.** Comparison of the TROPOMI/WFMD data to the TCCON based on daily means. Specified are the linear regression results and the correlation of the data sets, as well as the mean and standard deviation of the difference. To analyse the impact of outliers, the regression is also performed for the Huber linear regression model, which is robust to outliers.

In summary, the natural $XCH_4$ and XCO variations are well captured by the satellite data. We find a single measurement precision of the TROPOMI data of about $0.8\%$ for $XCH_4$ and $5.7\%$ for XCO, while the station-to-station accuracy of the satellite data is comparable to the TCCON.

## 4  Results

In this section we present first results from mission start until end of 2018. For temporally averaged data we grid the data on a $0.1° \times 0.1°$ grid instead of showing swath data, which is used for daily data and single satellite overpass detection. Before analysing the data more regionally, we want to provide a global overview.

The global distribution of retrieved XCO and $XCH_4$ for the year 2018 is shown in Figures 13 and 14, respectively. Clearly visible is the interhemispheric gradient with larger values on the northern hemisphere, where the majority of sources is located, for both data sets superimposed by enhancements over prominent source regions like anthropogenic emissions in China, India, and Southeast Asia.

Other visible XCO source regions include human-initiated biomass burning in Africa and South America for land clearing and land-use change, as well as wildfire emission in North America, which were exceptionally pronounced in 2018. The anthropogenic emissions of congested urban areas like Mexico City or Tehran are already unambiguously detected on such a global map without zooming in.

In the case of $XCH_4$ additional visible source regions apart from the anthropogenic sources in Asia, like fossil fuels or rice cultivation, include tropical wetlands as well as anthropogenic emissions in California or the Padan Plain in Italy. There is also





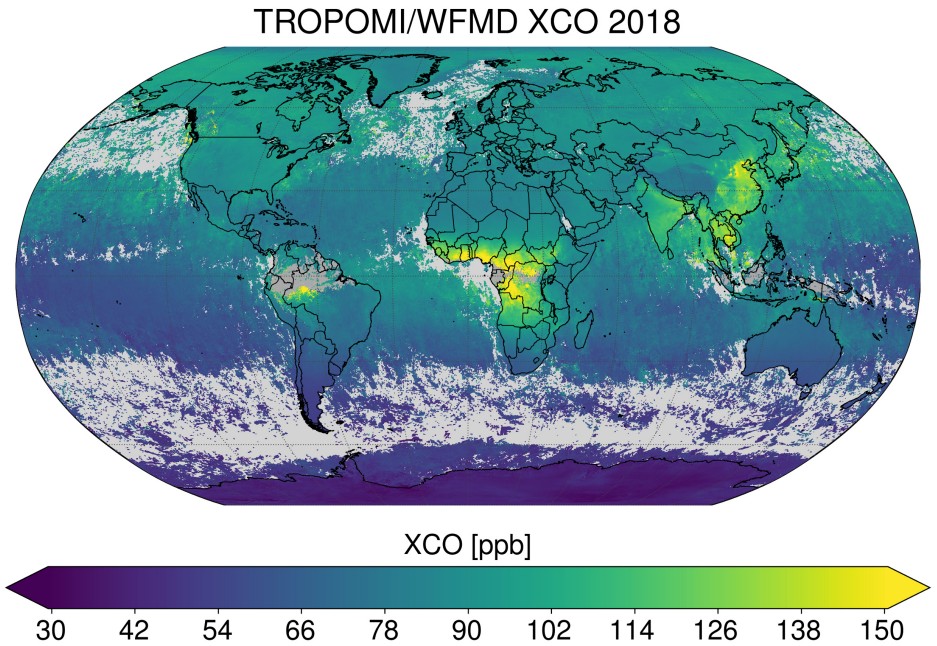

**Figure 13.** Global yearly average of TROPOMI/WFMD XCO for 2018.

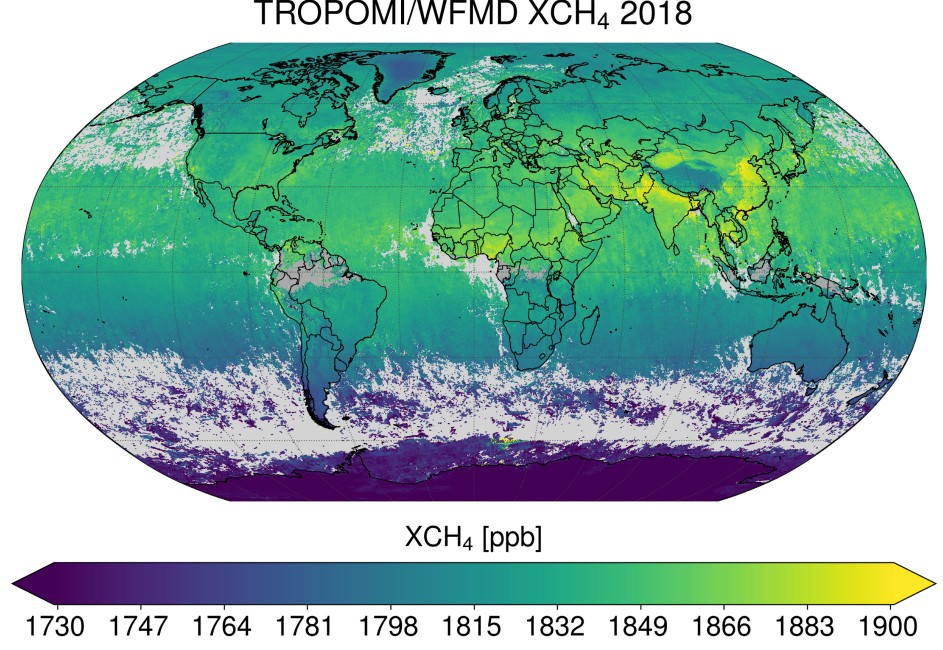

**Figure 14.** Global yearly average of TROPOMI/WFMD $XCH_4$ for 2018.



a distinct signal from the Etosha National Park in the north of Namibia containing significant areas of wetland like the Etosha Pan, a endorheic salt pan, which exhibits intermittent shallow inundation.

## 4.1 Comparison to operational products

The operational TROPOMI CO product is retrieved using the Shortwave Infrared Carbon Monoxide Retrieval (SICOR) algorithm (Landgraf et al., 2016) and the operational $CH_4$ product is based on RemoTeC (Hu et al., 2016), which is a physics-based approach originally developed for the $CO_2$ amd $CH_4$ retrievals from OCO and GOSAT. Although the operational algorithms and the scientific algorithm presented here use similar spectral bands, there are many differences concerning the details of each approach. For example, TROPOMI/WFMD is a weighted least squares approach, whereas SICOR and RemoTec are based on a Philips-Tikhonov regularisation scheme. There are also differences in the radiative transfer model, the quality filter, the used spectroscopy, and the state vector elements, in particular in the treatment of aerosols and clouds.

While WFMD and the operational $CH_4$ algorithm are mainly applicable to cloud-free scenes, the operational CO algorithm is designed to also handle cloudy observations under specific conditions. Both methane algorithms include a post-processing correction to improve the systematic albedo dependence. However, the details of this correction are again quite different: While the correction for the operational algorithm is based on linear regression relative to GOSAT retrievals, which are in turn bias-corrected against TCCON, the scientific WFMD algorithm uses a random forest regressor relative to a climatology as described in Section 2.5.3.

The comparisons are performed on a monthly basis with the latest version V01.02.02 of the operational products. Figure 15 shows the corresponding CO results for December 2018. The comparison of the global distribution of all quality filtered data for the respective algorithm illustrates that the spatial CO patterns are very similar for both algorithms. The operational CO algorithm exhibits a better coverage as it can handle a larger amount of cloudiness. For common scenes passing the quality filters of both algorithms the data sets are highly correlated with a correlation coefficient of $R = 0.98$ and also the regression slope is close to the 1:1 line confirming the good agreement. The mean bias between the two data sets is about $1\%$ and the standard deviation of the difference is comparable to the noise level. For the occasional very high CO abundances the results of the operational algorithm are a few percent larger than for WFMD, which is also reflected in a regression slope somewhat smaller than $1$.

The corresponding comparison of the $CH_4$ results is shown in Figures 16 and 17 for December and June 2018, respectively. WFMD exhibits a somewhat better coverage and also includes some retrievals over the ocean in contrast to the operational algorithm. Although, the prominent features like the interhemispheric gradient or source regions in Asia are similar and the correlation coefficients are close to $0.9$, the differences down to the last detail are more pronounced than for CO. This is reflected in both, the global maps and the scatter plots for common scenes. The global offset between the two data sets amounts to a few ppb and the standard deviation of the difference is again comparable to the noise level.

In December 2018 the methane abundances over the Bohai Economic Rim, including the cities of Beijing and Tianjin, are larger than in Southern China to the west of the Pearl River Delta for WFMD, whereas it is the other way round for the operational product, but this may be due to the different sampling. The $XCH_4$ distribution over the Sahara is more uniform for





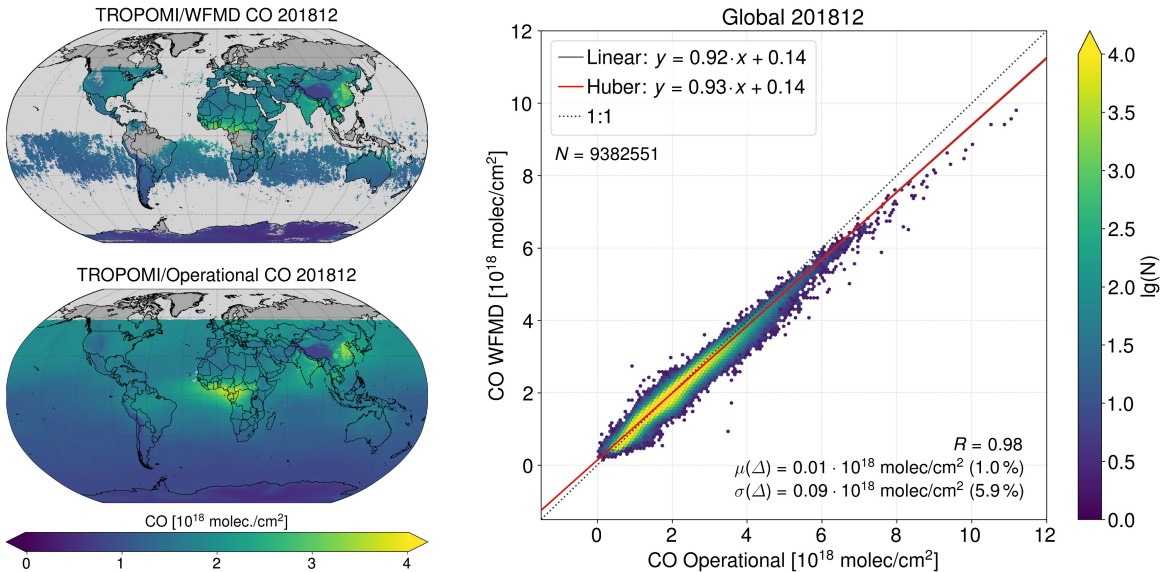

**Figure 15.** Comparison of TROPOMI/WFMD CO with the operational TROPOMI data for December 2018. The left hand panel depicts the global distribution of all quality filtered data for the respective algorithm. The right hand panel shows a scatter plot of all common scenes passing the quality filters of both algorithms summarising the linear regression results and the correlation of the data sets, as well as the mean and standard deviation of the difference.

WFMD, and the corresponding patterns of the operational product seem to vaguely resemble some albedo features with higher values over brighter parts of the Sahara. There is an obvious clustering of the common measurements around the 1:1 line, even for the largest values. Nevertheless, the linear regression line is somewhat distorted from the 1:1 line due to a slight shift of the two dominating densely populated sub-clusters.

5   In June 2018 there is a sharper $XCH_4$ gradient in the operational product when transitioning from the temperate into the low albedo boreal zone and the values over the boreal ecosystem are lower than for WFMD. In addition, there are enhanced methane abundances in the operational product over the Canadian province Nunavut in contrast to WFMD. Some occasional high values in the WFMD methane data over South America possibly attributable to surface roughness contribute to the rare outliers in the comparison scatter plot, which exhibits a clear linear relation close to the 1:1 line apart from that.

10   Overall, we find good agreement of our scientific CO product with the operational product based on the presented concise comparison. For $CH_4$ we find good agreement of the prominent features with some interesting differences in detail, including potential indications of residual albedo issues in the operational $XCH_4$ product. Further future analysis and understanding of the differences is believed to advance greenhouse gas retrievals from wide swath imaging satellites like TROPOMI under challenging conditions such as scenes with low surface reflectance or residual cloudiness.





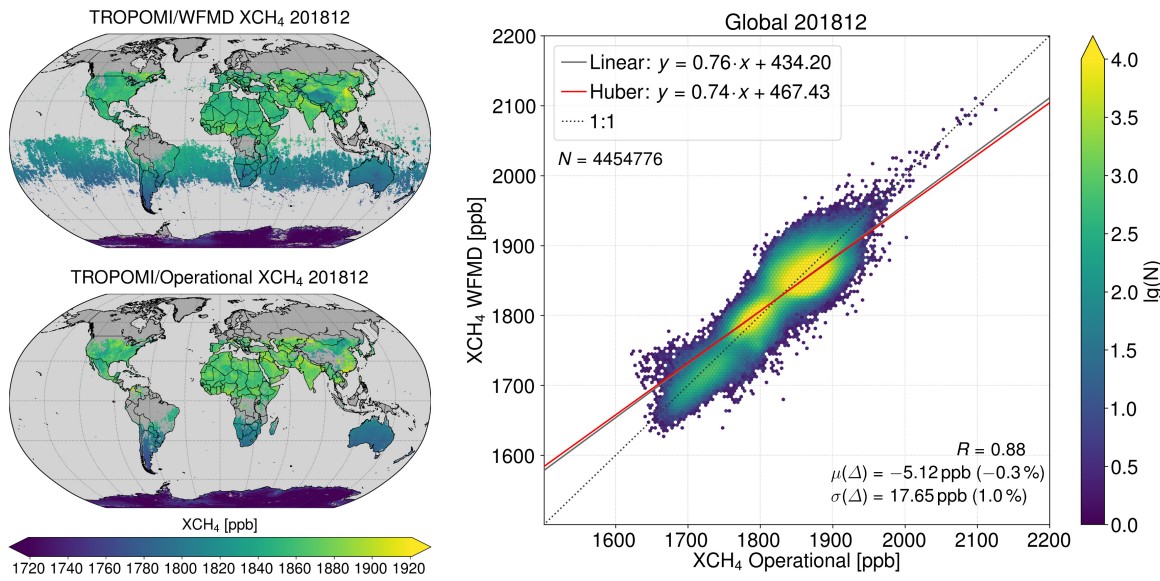

**Figure 16.** As Figure 15 but for methane.

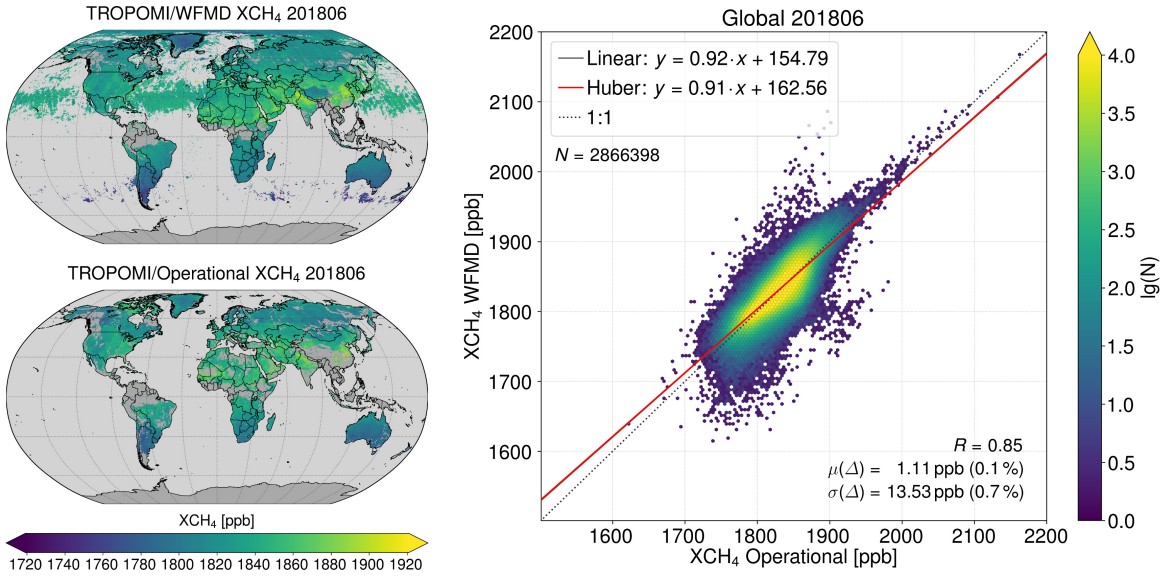

**Figure 17.** As Figure 16 but for June 2018.

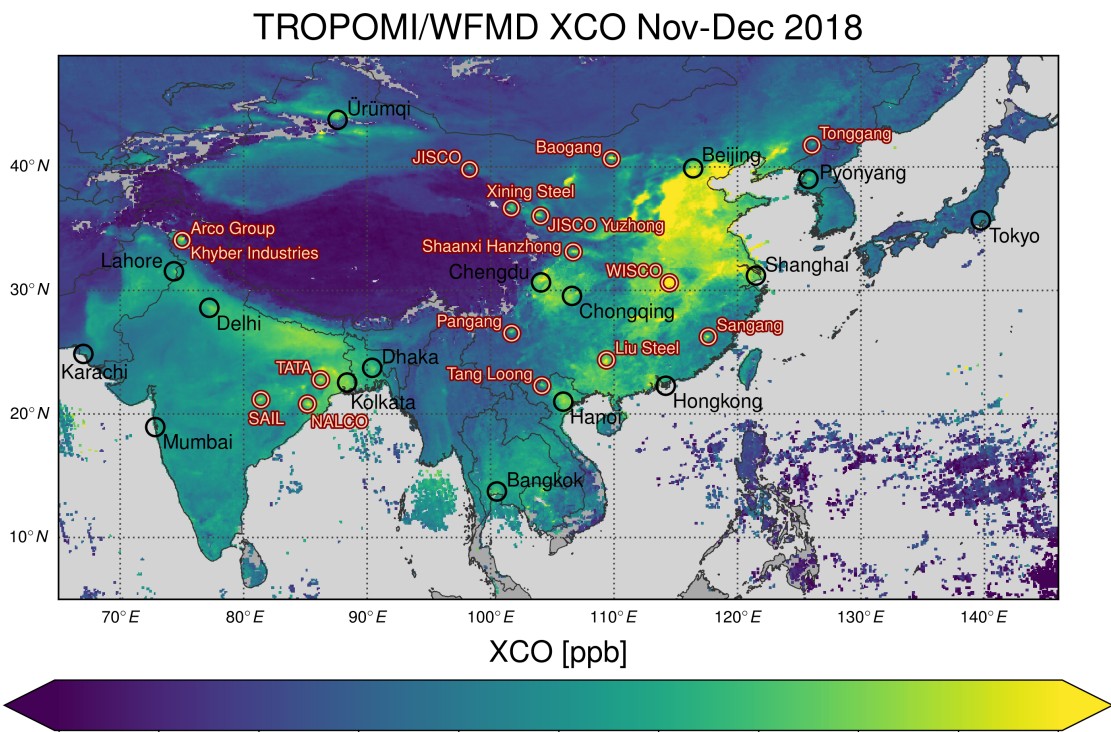

**Figure 18.** Carbon monoxide distribution for November and December 2018 over China, India, and Southeast Asia highlighting emissions of congested urban areas and industrial facilities.

## 4.2 Detection of emission sources

### 4.2.1 Carbon monoxide

Intense CO emissions of agglomeration areas, cities, and industrial facilities are clearly detected by TROPOMI. This is demonstrated using the example of China, India, and Southeast Asia in Figure 18. The two-month-average was chosen to get an overview of the complete region. Typically, larger emissions can even be detected in a single satellite overpass. The tracked facilities mainly belong to the Chinese and Indian iron and steel industry.

In steelmaking CO is formed during two processes. Firstly, it is an essential constituent of the blast furnace gas, which emerges when iron ore is reduced with coke to metallic pig iron. As the resulting pig iron has a relatively high carbon content, further processing is necessary to harden the metal. Therefore, the carbon-rich molten pig iron is converted to steel by lowering its carbon content via oxidation in the oxygen converter process (Linz-Donawitz-steelmaking). The resulting converter gas predominantly consists of CO ($> 60\%$).

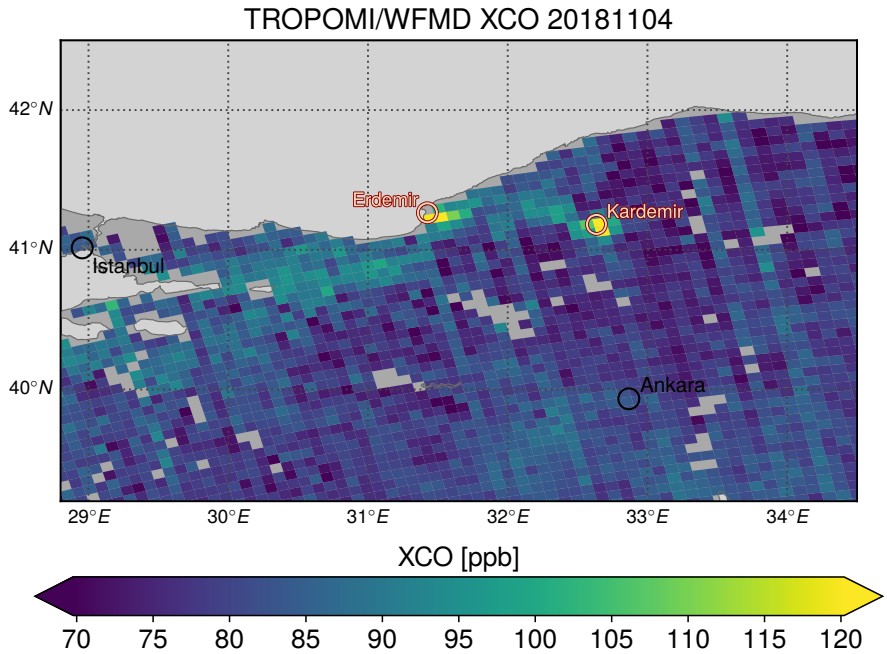

**Figure 19.** Carbon monoxide enhancement due to emissions from steel plants in Turkey.

The detected factories include steel plants of Baotou Iron & Steel Group (Baogang), Wuhan Iron & Steel Co. (WISCO), Panzhihua Iron & Steel Group (Pangang), Fujian Sangang Group (Sangang), Liuzhou Iron & Steel Co. (Liu Steel), Jiuquan Iron & Steel Co. (JISCO), Xining Special Steel, Shaanxi Hanzhong Iron & Steel Co., Tonghua Iron & Steel Group (Tonggang), Steel Authority of India Limited (SAIL), and TATA Steel. Further emitters include other metal processing industries such as

the National Aluminium Company (NALCO), the Tang Loong Industrial Park in Vietnam, and the cement production plants of the Arco Group and Khyber Industries in the Kashmir Valley, where the CO accumulates between the mountain ranges.

CO emissions from the steel industry can also be observed in other regions of the world, for example in Turkey. Figure 19 shows that two of the largest steel plants of the country are detected in a single satellite overpass. The plants are operated by the Turkish steel producers Ereğli Demir ve Çelik Fabrikaları (Erdemir) and Karabük Demir Çelik Fabrikaları (Kardemir) with

yearly production capacities of about 3 million t crude steel each. It can also be seen that the CO product exhibits striping in flight direction for single overpasses similar to the operational product (Borsdorff et al., 2018).

There are also examples of detected CO emissions from steel works in Europe. Figure 20 illustrates such a case and shows that CO emissions from the largest steel plant in Poland operated by ArcelorMittal in the industrial city Dąbrowa Górnicza in the Upper Silesian metropolitan area are detected in a single overpass. As can be seen, the corresponding pronounced plume

coincides with the boundary layer wind direction and the striping is observable as well.

Another prominent source of CO is fire. In September 2018 a peat bog on the military training area WTD 91 in the Emsland region was accidentally set on fire by the German army and burnt several weeks. The corresponding CO plume is clearly

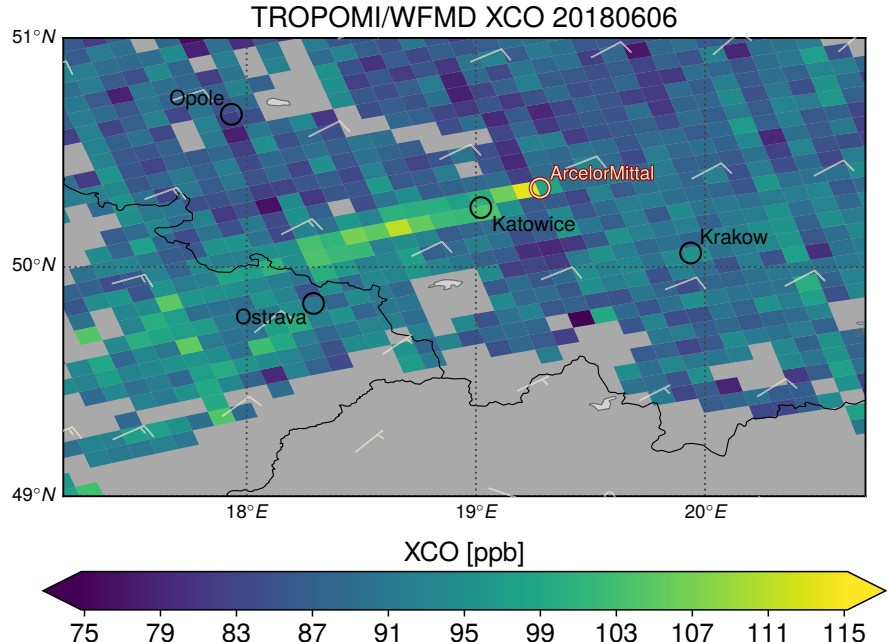

**Figure 20.** Carbon monoxide enhancement due to emissions from the ArcelorMittal steel plant in Dąbrowa Górnicza in the Upper Silesian metropolitan area in Poland. Also shown is the mean wind in the boundary layer obtained from ECMWF data.

detected and aligns well with the wind direction (Figure 21). The scenes right above the origin of the fire are automatically excluded by the quality filter because of the strong formation of smoke potentially shielding the subjacent partial columns similar to thick clouds.

The difference between smoke and clouds is the particle size distribution. While clouds consist of water droplets with an
effective radius of about $10\,\mu m$, the mass distribution of smoke plumes shows a prominent peak at about $0.3\,\mu m$ (Stith et al., 1981) but is nevertheless dominated by a small number of supermicron-sized particles (Radke et al., 1990). The submicron particles reduce the visibility and lead to an extended smoke plume over large distances in the true color reflectances from VIIRS shown in Figure 21. However, these small particles are not a major issue for the satellite measurements taken at $2.3\,\mu m$. The satellite retrievals near the origin of the fire are rather affected by the large supermicron-sized particles, which are getting
more and more negligible when departing from the seat of fire due to their rapid fallout. This is the reason that in sufficient distance from the fire the corresponding measurements pass the quality filter despite efficient scattering in the visible spectral range manifesting in an extensive plume in the VIIRS image. On the other hand, even very small clouds, which are barely visible in the VIIRS image at this resolution, are rigorously filtered out. This indicates that the algorithm implicitly distinguishes between smoke and clouds according to their particle sizes and that a reliable CO retrieval is possible in smoke plumes in the
far field of the fire origin. A thorough discussion of the sensitivity of CO measurements in conjunction with smoke from fires can be found in the revised version of Schneising et al. (2019).



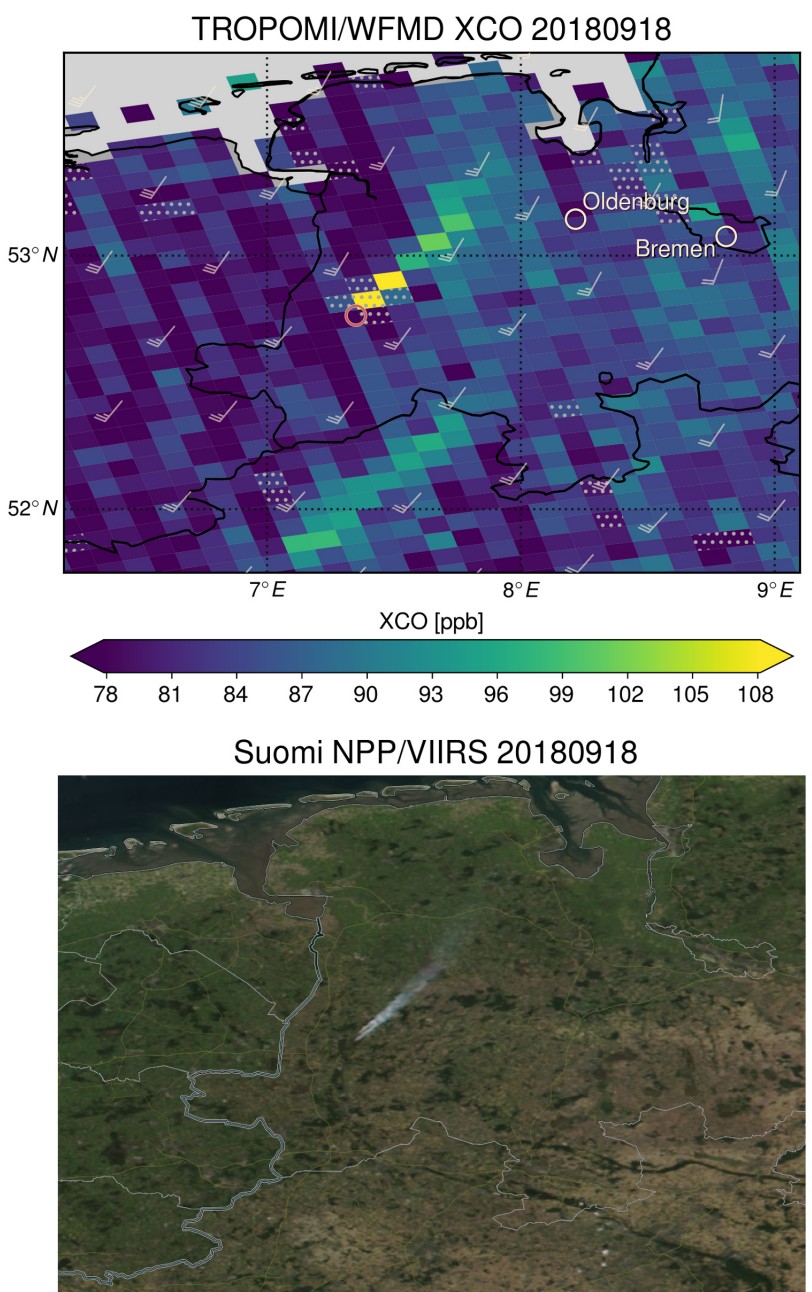

**Figure 21.** Carbon monoxide enhancement due to the peat fire (red circle) in the Emsland region in Germany. Dotted scenes are excluded by the quality filter. Also shown is the boundary layer wind from ECMWF. The bottom panel shows the corresponding true colour reflectances from the Visible Infrared Imaging Radiometer Suite (VIIRS).

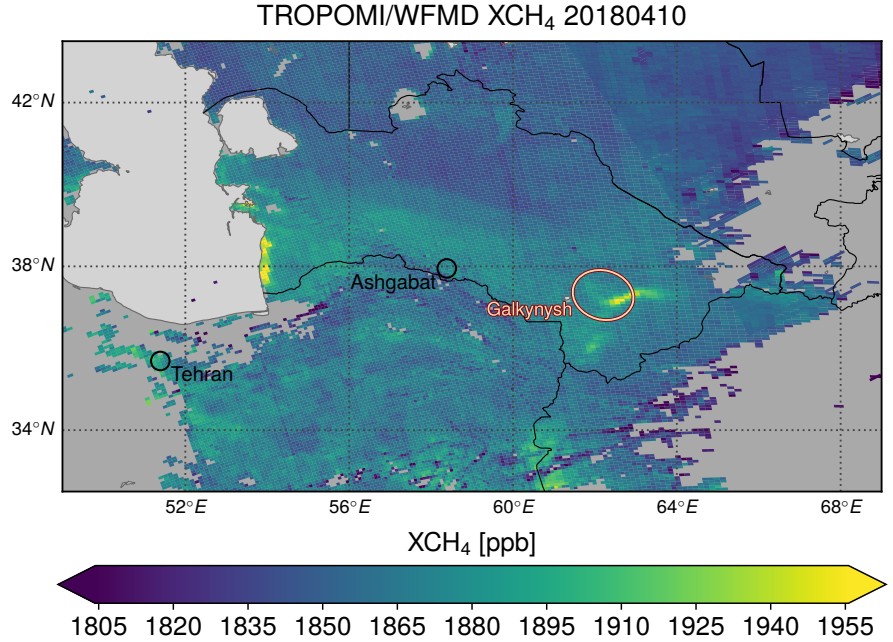

**Figure 22.** Methane enhancement due to emissions from the world's second-largest natural gas field Galkynysh in Turkmenistan.

The total column enhancement $E$ relative to background values allows to roughly estimate the emitted mass flux $\Phi$ of CO from the mean boundary layer wind speed $v$ and the plume width $x_\perp$ perpendicular to the wind direction only using measurements passing the quality filter:

$$\Phi = E \cdot v \cdot x_\perp \tag{9}$$

With $E = (7 \pm 0.7) \cdot 10^{17}\,\mathrm{molec/cm^2}$, $v = 12 \pm 3\,\mathrm{m/s}$, and $x_\perp = 5 \pm 1\,\mathrm{km}$, the emission on 18 September amounts to about $\Phi = 1.7 \pm 0.6\,\mathrm{ktCO}$. According to Kohlenberg et al. (2018), a $CO{:}CO_2$-emission factor of $16 \pm 3\,\%$ for boreal peat fires is assumed implying an associated $CO_2$ emission of approximately $10.5 \pm 4.0\,\mathrm{ktCO_2}$ on that day. Compared to the German yearly total budget of about $800\,\mathrm{MtCO_2/yr}$ the emissions of the Emsland peat fire are small even if one assumes that the fire burnt several weeks at this strength.

**4.2.2   Methane**

One integral component of anthropogenic methane sources is emissions from the energy sector. As an example of methane leakage from natural gas production, Figure 22 shows that the emissions of the world's second-largest natural gas field Galkynysh in Turkmenistan, which is operated by Türkmengaz, can be clearly detected in a single satellite overpass. Also visible are $XCH_4$ enhancements over the productive South Caspian oil and gas basins, the oil and gas infrastructure at the Turkmen coast
of the Caspian Sea, and smaller oil and gas fields south of Galkynysh.




Emissions from oil and gas production are important to monitor, because methane leaks offset the climate change benefits of natural gas or oil over coal if the leakage exceeds a certain threshold (Alvarez et al., 2012; Farquharson et al., 2016). There are several studies suggesting that the oil and gas industry leaks more methane than assumed in inventories, at least locally or temporally (Brandt et al., 2014; Schneising et al., 2014; Alvarez et al., 2018) and the potential heterogeneity among the sector

complicates the specification of typical emission rates.

Another source region is the Central Valley in California with combined anthropogenic emissions from oil fields and agriculture (see Figure 23). While one main area of oil production is located in Kern County around Bakersfield, the dairy and cattle industry extends more or less over the whole valley with largest livestock density in the counties San Joaquin, Stanislaus, Merced, Kings, and Tulare (Mauger et al., 2015). A reliable disentanglement of the emissions from the oil and agriculture sec-

tors requires exact knowledge of the meteorology and unmistaken prior knowledge of the distribution of the different source types or methane isotopologue information, which is however not yet available from satellite observations. As already mentioned in Section 3, the $100\,\mathrm{km}$ collocation radius standardly used in the validation intersects the Kern county source region for the Edwards TCCON site. As a consequence, the collocation radius is reduced to $50\,\mathrm{km}$ for Edwards to ensure the representativity of the satellite measurements used in the validation.

The two hitherto presented methane source regions Turkmenistan and the Central Valley in California, which are both detected in a single TROPOMI overpass, were already identified in yearly averages of SCIAMACHY data (Buchwitz et al., 2017).

An additional source of methane from the energy sector is emissions from coal mining. The physical process of coal extraction directly releases methane, which was previously trapped within the coalbed in the form of gas particles adsorbed at

coal grains. For reasons of safety, the coal mine methane is diluted with air below the explosive range and released through ventilation shafts to the surface.

Poland's primary energy consumption and electrical power generation relies strongly on coal, which helped the country to achieve one of the lowest energy import dependencies in the European Union (European Statistical Office, 2018), which is measured by the share of net imports in gross inland energy consumption (sum of energy produced and net imports). The

energy dependence rate of Poland was about $30\,\%$ in 2016 compared to an EU-wide average of $54\,\%$ meaning that the majority of the EU's energy needs are met by net imports. Only three EU countries have a lower energy dependence rate than Poland, namely Romania, Denmark, and Estonia.

Poland is the largest coal-mining country in Europe and among the top ten coal producers in the world (BP, 2018) with huge reserves of hard coal and lignite (Polish Geological Institute, 2018a, b). The major coal basin is the Upper Silesian Coal Basin

(USCB), which is larger than $5000\,\mathrm{km}^2$ and hosts $80\,\%$ of the anticipated domestic hard coal resources. All operating hard coal mines of the country are situated in the USCB except the Bogdanka Mine in the Lublin Coal Basin in the east of Poland. The USCB is shown in Figure 24 highlighting individual mines. The corresponding methane plume is vaguely perceptible and coincides with the wind direction. This is an example of emissions, which are obviously close to the detection limit for a single overpass.

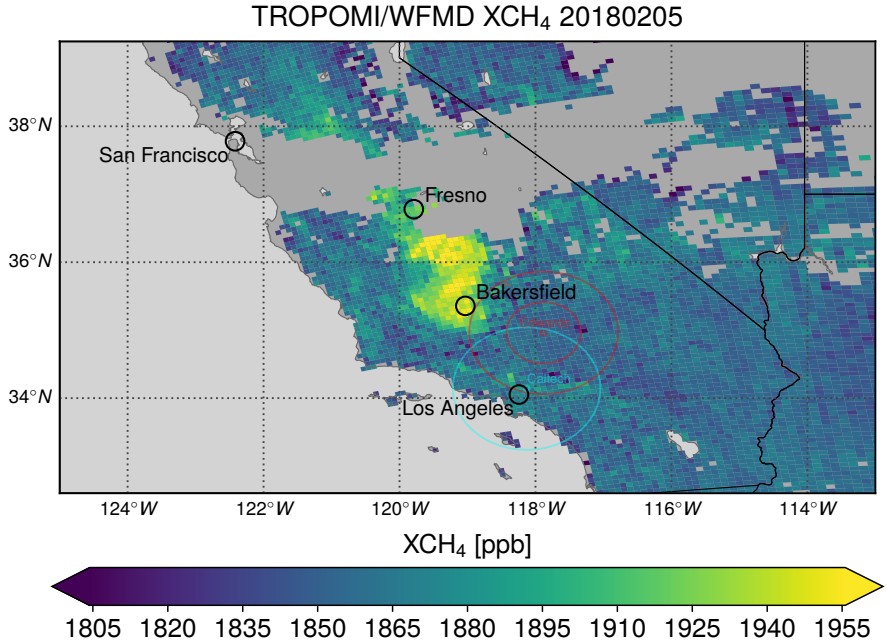

**Figure 23.** Methane enhancement due to anthropogenic emissions from the Californian Central Valley. In contrast to the Caltech (and JPL) TCCON site (shown in cyan), the source region is intersected by the $100\,\mathrm{km}$ collocation radius standardly used in the validation for the Edwards site (red). Therefore, the collocation radius is reduced to $50\,\mathrm{km}$ for Edwards. Due to the altitude representativity criterion, different air masses are analysed in the validation at Caltech/JPL and Edwards, although the corresponding collocation circles overlap. For a discussion of the collocation criteria at Edwards and Caltech/JPL see also Section 3.

## 5 Conclusions

We have introduced a scientific algorithm to retrieve XCO and $XCH_4$ simultaneously from shortwave infrared spectra recorded by the TROPOMI instrument onboard the Sentinel-5 Precursor satellite. The error analysis based on synthetic data and the successful validation with independent reference data from the TCCON has demonstrated that the algorithm is suitable to

5     retrieve XCO and $XCH_4$ from real TROPOMI data well within the mission requirements after quality filtering (see Table 4). The corresponding quality filter is based on a machine learning approach utilising a random forest classifier. As cloud data from VIIRS onboard Suomi NPP was only used in the preceding supervised learning process and is no longer needed in the actual quality prediction of individual previously unseen measurements after completion of the training, the quality filter is independent of the continuous availability of external cloud information. The performance of the retrieval algorithm is expected

10     to further improve in the future, for example with respect to striping, due to a refined calibration of the TROPOMI instrument and/or dedicated algorithm advancements.

The good global agreement of our scientific products with the operational products for the analysed example cases further underlines the quality of the presented algorithm. The differences in detail for $XCH_4$ can be thought of as a stimulation





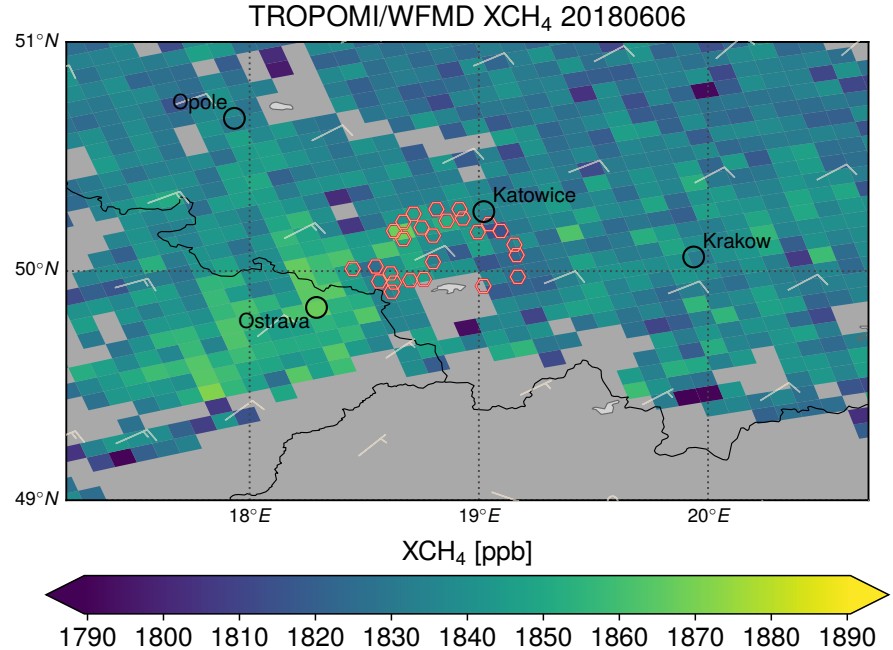

**Figure 24.** Methane distribution over the Upper Silesian Coal Basin in Poland. Individual mines are highlighted by red hexagons.

**Table 4.** Achieved product quality compared to the mission requirements extracted from ESA (2017).

| Data product | Vertical resolution | Bias [%] | | Random [%] | |
|---|---|---|---|---|---|
| | | required | achieved | required | achieved |
| Carbon monoxide (CO) | Total column | 15 | 2.1 | 10 | 5.7 |
| Methane ($CH_4$) | Total column | 1.5 | 0.2 | 1.0 | 0.8 |

for further future analysis. The understanding of these differences will likely allow to symbiotically advance both retrieval algorithms under challenging conditions such as scenes with low surface reflectance or residual cloudiness. Moreover, the scientific and operational products are predestined to be used together with other products in an ensemble approach to benefit from the large range of respective realisations of different physical aspects in the individual retrieval algorithms.

5    Nevertheless, the results of the presented scientific algorithm are also valuable in its own right as TROPOMI enables the determination of XCO and $XCH_4$ with unprecedented level of detail on a global scale introducing new areas of application. It was shown that CO emissions of agglomeration areas, industrial facilities, in particular from the steel industry, and fires are readily detected, often even in a single satellite overpass. The same is true for $CH_4$ emissions from the energy sector including leakage from oil and gas production and coalbed methane from coal mining. The future quantitative reinforcement

10   of these primarily qualitative findings will potentially enable emission monitoring and air quality assessments, ideally on a



daily recurrent basis. Furthermore, improved knowledge of the methane cycle, which is essential for better prediction of future climate, can be derived by combining inverse modelling with a comprehensive monitoring system comprising complementary information from accurate ground-based in-situ measurements and satellite observations with an unique combination of high precision, spatiotemporal resolution, and global coverage.

*Data availability.*    The carbon monoxide and methane data sets presented in this publication can be accessed via http://www.iup.uni-bremen. de/carbon_ghg/products/tropomi_wfmd/.

*Author contributions.*    OS: writing the paper, design and operation of the TROPOMI/WFMD satellite retrievals, data analysis, interpretation. MB, MR, HB, JPB: significant conceptual input to writing, design of the TROPOMI/WFMD satellite retrievals, interpretation. TB, JL: design and execution of the operational TROPOMI CO and CH$_4$ satellite retrievals, interpretation. NMD, DGF, DWTG, FH, CH, LTI, RK, IM, JN,
CP, DFP, SR, KSh, KSt, RS, VAV, TW, DW: operation of the TCCON retrievals for the various sites, interpretation. All authors discussed the results and commented on the manuscript.

*Competing interests.*    The authors declare that they have no conflict of interest.

*Acknowledgements.*    This publication contains modified Copernicus Sentinel data (2017,2018). Sentinel-5 Precursor is an ESA mission implemented on behalf of the European Commission. The TROPOMI payload is a joint development by ESA and the Netherlands Space
Office (NSO). The Sentinel-5 Precursor ground-segment development has been funded by ESA and with national contributions from The Netherlands, Germany, and Belgium. The research leading to the presented results has in part been funded by the ESA projects GHG-CCI, GHG-CCI+, and S5L2PP, the Federal Ministry of Education and Research project AIRSPACE, and by the State and the University of Bremen.
     TCCON data were obtained from the TCCON Data Archive, hosted by CaltechDATA, California Institute of Technology (https://tccondata. org/) extended by updates for Eureka, Ny-Ålesund, Białystok, Orléans, Burgos, Darwin, and Wollongong provided by the respective PIs of
these sites. We thank Coleen Roehl and Paul Wennberg of Caltech as well as Jean-François Blavier and Geoffrey Toon of JPL for their efforts to operate the TCCON sites at Park Falls, Lamont, JPL, and Caltech and for providing support for the entire TCCON, which has evolved to the primary validation resource for our satellite data sets. The East Trout Lake TCCON station is supported by the Canada Foundation for Innovation, the Ontario Research Fund, Environment and Climate Change Canada (ECCC), and the Canadian Space Agency (CSA). The Eureka TCCON measurements were made at the Polar Environment Atmospheric Research Laboratory (PEARL) by the Canadian Network
for the Detection of Atmospheric Change (CANDAC), primarily supported by the Natural Sciences and Engineering Research Council of Canada (NSERC), ECCC, and CSA. The US TCCON sites used in this analysis are supported by the NASA OCO-2 Project and NASA Carbon Cycle Science Program. The TCCON sites at Tsukuba and Burgos are supported in part by the GOSAT series project. Site support for Burgos is provided by the Energy Development Corporation (EDC, Philippines). The Ascension Island TCCON station has been supported by the European Space Agency (ESA) under grant 3-14737 and by the German Bundesministerium für Wirtschaft und Energie (BMWi)



under grant 50EE1711C. The Lauder TCCON programme is core-funded by NIWA through New Zealand's Ministry of Business, Innovation and Employment. JN, CP, and TW acknowledge financial support by the DFG within the Transregio (AC)[3].

We acknowledge the use of VIIRS imagery from the NASA Worldview application (https://worldview.earthdata.nasa.gov/) operated by the NASA/Goddard Space Flight Center Earth Science Data and Information System (ESDIS) project. We also thank the European Centre
5  for Medium-Range Weather Forecasts (ECMWF) for providing the meteorological analysis.



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
