# Peer review of "A scientific algorithm to simultaneously retrieve carbon monoxide and methane from TROPOMI onboard Sentinel-5 Precursor"

_Atmospheric Measurement Techniques, 2019_

## Referee Comment (RC1) · Anonymous Referee #1 · 10 Aug 2019

General comments :

The paper describes the first results of an algorithm to simultaneously retrieve CO and CH4 from TROPOMI radiance measurements using an error analysis based on synthetic data and a cloud filter based on a machine learning for XCH4 data. The paper is well written with a clear presentation of the retrieval algorithm as well as a careful consideration and analysis of the results and their validation. I only have a few minor remarks and questions. The paper is interesting for the scientific community of TROPOMI data, CO and CH4 retrievals and should be published in AMT.

Specific comments :

[Figure]

- p.4. line 15 : Details of the goal and the contribution of this new algorithm are missing in the introduction. Also, as you mentioned in your abstract EGU 2019, you should said something similar about "We present first results for both CO and CH4 trace gases obtained using the new version v1.2 of the scientific retrieval algorithm WFM-DOAS".

- p.6, line 9 : "the look-up table is only covering direct nadir conditions". Why ?

- p.8, line 10 : The scenario class of profiles are extracted from the radiative transfer model MODTRAN. Why from this RTM ?

- p.8, line 11 : "which differ from the US Standard Atmosphere" but how much ?

- p.13, line 20 : "20% of the training data are randomly drawn and retained as test data". How did you choose 20 % ?

- p.31, line 5 : how did you measure the plume width perpendicular to the wind direction ?

Technical comments :

- p.2, line 19 : "Moreover, it" -> "Moreover, CO" as you are talking about O3 just before...

- p.3, line 22 : I would add a reference on this sentence about the high precision of TROPOMI.

- p.4, line 18 : is a linear-least squares method

- p.4, line 19 : vertical profiles of trace gases.

- p.5, line 27 : where T is the matrix transpose.

- p.8, line 14 : acronym of ASTER and USGS should be mentioned here and not after.

- p.11, line 11 : have also to be implemented -> have to be implemented

- p.11, figure 6 : what is the meaning of the label for the surface type (range from 1 to 20). I did not find the information in the references mentioned in the paragraph 2.4.

- p.13, line 12 : acronym of VIIRS is defined pages 15 and 30 but must be mentioned here too

- p.27, line 11 : do you have a reference for this percentage ?

- p.33, figure 23 : we do not see very well the cyan color on this figure. Same case for "Edwards" in red letters.

---

## Referee Comment (RC2) · Anonymous Referee #2 · 5 Sep 2019

Review of AMT-2019-243

A scientific algorithm to simultaneously retrieve carbon monoxide and methane from TROPOMI onboard Sentinel-5 Precursor

By Oliver Schneising, Michael Buchwitz, Maximilian Reuter, Heinrich Bovensmann, John P. Burrows, Tobias Borsdorff, Nicholas M. Deutscher, Dietrich G. Feist, David W. T. Griffith, Frank Hase, Christian Hermans, Laura T. Iraci, Rigel Kivi, Jochen Landgraf, Isamu Morino, Justus Notholt, Christof Petri, David F. Pollard, Sébastien Roche, Kei Shiomi, Kimberly Strong, Ralf Sussmann, Voltaire A. Velazco, Thorsten Warneke, and Debra Wunch

**General comments.**

  The authors present a scientific algorithm to simultaneously retrieve carbon monoxide and methane from TROPOMI onboard Sntinel-5 Precursor. I understand that this activity is important to calibration and validate both of TROPOMI spectra and operational products. However, it is unclear that what is the object for developing the proposed algorithm or how is the difference between the proposed algorithm and the operational one. In this paper, a lot of demonstrated results are described. Unfortunately, it is hard to understand the usefulness of this algorithm. It is easy to understand, if this paper focus on the validation of TROPOMI operational products. Several topics are described in this paper but the relevance between these topics are poor. Of course, the individual topics are important. So, I recommend the authors will reconstruct the frame of this paper.

  For these reason, I recommend this paper for publication with major changes to the technical content.

**Specific comments.**

**Abstract**

1. Describe the character of proposed science algorithm. Especially, the comparison between the proposed algorithm and the operational one.
2. Describe the motivation or object for developing proposed algorithm.
3. Describe a full word of "DOAS".

4. Page 2, line10: What is a "reference data".

5. Page 2, line13: Why emission sources have to be identified? Describe the object or back ground.

6. I understand that one of target for developing this scientific algorithm is to validate the operational TROPOMI XCH4 and XCO products. If so, it might be described in.

**1. Introduction**

7. Page 4, line 9 to line15: It is unclear what is the requirement of a scientific algorithm ? I understand that validation of operational products, calibration of TROPOMI spectra, and reduce the random and systematic error of XCH4 and XCO with scientific algorithm are first objects. Second is new findings with scientific algorithm. If so, describe more clearly.

**2. WFM-DOAS retrieval algorithm**

8. Figure 1, It is unclear the coverage of gray hatch. Add table for these coverages.

9. Table 2, Describe the meaning of "T", "p", "BL", "R", " $\tau$ ".

10. Figure 5, Describe the full word of "cum", "cir"

11. Figure 6, Describe the meaning of numbers for surface type.

12. Page 12, line 8 to 15: In the other algorithm to retrieving the XCH4 and XCO used O2A spectra, to identify the photon path with precisely. However, this algorithm is not employed the O2A spectra to identify the photon path. Instead of O2A spectra, this algorithm used the ECMWF-Based mole fraction computation. The authors are concluded that the proposed algorithm is more faster and accurate than that of O2A based processing system. However, it is not quantitative. The authors have to assess more quantitatively.

13. Page 13, line 22: Make table for "all 25 features".

14. Page 15, Figure 8: What is the meaning of "QUAL=1"

**4. Results**

15. Page 24, line 4: Correct the capital position.

16. Page 24, line 6: typo "amd".

17. Page 25, Figure 15: Why the yield rate for XCH4 is drastically different between WFMD CO and Operational CO?

**4.2 Detection of emission sources**

18. Page 27, Figure 18: How is the operational products? Is it possible to identify the emission sources with operational products?

19. Figure 19 to figure 22 are almost same information. These figures are just illustration. Make more clear sentence.

**5. Conclusions**

20. Page 33, line 10, The sentence " for example with respect to striping," is not touched on this paper. Adding the reference or explanation

End of document

---

## Author Comment (AC1) · 2 Oct 2019

**Final response to referee comments on paper amt-2019-243**

First of all, we would like to thank reviewer #1 for his/her constructive comments, which helped to improve the manuscript. Below we give answers and clarifications to all comments made by the referee (repeated in italics).

**Anonymous Referee #1**

**Specific comments**

**Reviewer:** p.4. line 15: Details of the goal and the contribution of this new algorithm are missing in the introduction. Also, as you mentioned in your abstract EGU 2019, you should said something similar about "We present first results for both CO and CH4 trace gases obtained using the new version v1.2 of the scientific retrieval algorithm WFM-DOAS".

Authors: We have modified the introduction accordingly. The last paragraph of the introduction now reads: "Here we introduce a scientific algorithm to retrieve CO and  $CH_4$ simultaneously from TROPOMI that has the objective to complement the operational algorithms in the sense described above and to provide new geophysical insights, whilst performing within the mission requirements concerning random and systematic errors at the same time. The presented scientific algorithm differs from the operational algorithms in several respects (Landgraf et al., 2016; Hu et al., 2016) (see also Section 4.1 for a summary of the differences) and the corresponding products are thus predestined to be used together with the operational products in an ensemble approach. After a thorough description of the algorithm including error characteristics based on synthetic data and validation with independent reference data, we present first results of our new algorithm for both trace gases demonstrating the broad consistency with the operational products for example cases and the potential to advance the new application fields, for which TROPOMI's groundbreaking features pave the way."

**Reviewer:** p.6, line 9: "the look-up table is only covering direct nadir conditions". Why?

Authors: The explanation is added to the manuscript, namely to limit the dimension of the look-up table to a reasonable size. With n sampling points for the viewing zenith angle, the look-up table size would increase by a factor of n.

**Reviewer:** p.8, line 10 : The scenario class of profiles are extracted from the radiative transfer model MODTRAN. Why from this RTM ?

Authors: We have modified this text passage to better represent the origin of the profiles. Anderson et al. (1986) have designed this data base to be incorporated in radiative transfer models in general. It was chosen because the included profiles are considered realistic as they are based on measurements and theoretical predictions. The passage now reads: "In order to examine the sensitivity to vertical profile variations, the scenario class of *Profiles* includes several realistic model atmospheres based on measurements and theoretical predictions (Anderson et al., 1986), with all methane profiles scaled to have surface values of 1850 ppb in each case to better represent current atmospheric conditions."

Reviewer: p.8, line 11 : "which differ from the US Standard Atmosphere" but how much ?

Authors: A visualisation of the different vertical profiles can be found in Appendix A of Anderson et al. (1986). This information is added to the manuscript.

**Reviewer:** p.13, line 20: "20% of the training data are randomly drawn and retained as test data". How did you choose 20%?

Authors: A split with 20% test data is widely used. This is explained in the revised version with two example references (Suthaharan, 2016; Hino et al., 2018).

**Reviewer:** p.31, line 5 : how did you measure the plume width perpendicular to the wind direction ?

**Authors:** The estimation of the plume width is described in more detail in the revised version. The corresponding text passage now reads: "The plume width is on the same order of magnitude as the instrument's spatial resolution and the enhancement is thus calculated for the plume scene passing the quality filter which is nearest to the fire origin. As the wind direction is approximately perpendicular to one of the scene diagonals, the corresponding plume width  $x_{\perp}$  is estimated by  $\frac{a}{\sqrt{2}}$  assuming a quadratic scene with a side length a of about 7 km."

**Technical comments**

**Reviewer:** p.2, line 19 : "Moreover, it"  $\rightarrow$  "Moreover, CO" as you are talking about O3 just before...

Authors: Has been changed in the revised version.

**Reviewer:** p.3, line 22 : I would add a reference on this sentence about the high precision of TROPOMI.

Authors: The reference for the whole paragraph including the high precision statement is the cited paper Veefkind et al. (2012).

**Reviewer:** p.4, line 18 : is a linear-least squares method

Authors: Has been changed in the revised version.

**Reviewer:** p.4, line 19 : vertical profiles of trace gases.

Authors: This sentence shall also comprise temperature and pressure profiles. Therefore, the original phrasing was retained unchanged.

**Reviewer:** p.5, line 27 : where T is the matrix transpose.

Authors: Has been added in the revised version.

**Reviewer:** p.8, line 14 : acronym of ASTER and USGS should be mentioned here and not after.

Authors: The acronyms are explained at this place in the revised version.

**Reviewer:** p.11, line 11 : have also to be implemented  $\rightarrow$  have to be implemented

Authors: Has been changed in the revised version.

**Reviewer:** p.11, figure 6 : what is the meaning of the label for the surface type (range from 1 to 20). I did not find the information in the references mentioned in the paragraph 2.4.

**Authors:** The labels are explained in Appendix 6 of the Global Land Cover Characterization (GLCC) readme file. The meaning of the labels has been added to the figure caption in the revised version.

**Reviewer:** p.13, line 12 : acronym of VIIRS is defined pages 15 and 30 but must be mentioned here too

Authors: Has been done in the revised version.

**Reviewer:** p.27, line 11 : do you have a reference for this percentage ?

Authors: We have added two references in the revised version. The sentence now reads: "The resulting converter gas predominantly consists of CO ( $\approx 70\%$ ) (Ishioka et al., 1992; CarbonNext, 2017).

**Reviewer:** p.33, figure 23 : we do not see very well the cyan color on this figure. Same case for "Edwards" in red letters.

Authors: The colours have been changed in the revised version for the sake of better visibility.

**References**

Anderson, G. P., Clough, S. A., Kneizys, F. X., Chetwynd, J. H., and Shettle, E. P.: AFGL Atmospheric Constituent Profiles (0-120 km), Environmental Research Papers, NO. 954, AFGL-TR-86-0110, https://apps.dtic.mil/dtic/tr/fulltext/u2/a175173.pdf, 1986.

CarbonNext: Map of relevant CO2 and CO containing gases, http://carbonnext.eu/Deliverables/ \_/D1.1MapofrelevantCO2andCOcontaininggases.pdf, 2017.

Hino, M., Benami, E., and Brooks, N.: Machine learning for environmental monitoring, Nature Sustainability, 1, 583–588, https://doi.org/10.1038/s41893-018-0142-9, 2018.

Ishioka, M., Okada, T., and Matsubara, K.: Formation and characteristics of vapor grown carbon fibers prepared in Linz-Donawitz converter gas, Carbon, 30, 975–979, https://doi.org/10.1016/0008-6223(92)90124-F, 1992.

Suthaharan, S.: Machine Learning Models and Algorithms for Big Data Classification, Springer, New York, 2016.

Veefkind, J. P., Aben, I., McMullan, K., Förster, H., de Vries, J., Otter, G., Claas, J., Eskes, H. J., de Haan, J. F., Kleipool, Q., van Weele, M., Hasekamp, O., Hoogeveen, R., Landgraf, J., Snel, R., Tol, P., Ingmann, P., Voors, R., Kruizinga, B., Vink, R., Visser, H., and Levelt, P. F.: TROPOMI on the ESA Sentinel-5 Precursor: A GMES mission for global observations of the atmospheric composition for climate, air quality and ozone layer applications, Remote Sensing of Environment, 120, 70–83, https://doi.org/10.1016/j.rse.2011.09.027, 2012.

---

## Author Comment (AC2) · 2 Oct 2019

**Final response to referee comments on paper amt-2019-243**

First of all, we would like to thank reviewer #2 for his/her constructive comments, which helped to improve the manuscript. Below we give answers and clarifications to all comments made by the referee (repeated in italics).

**Anonymous Referee #2**

**General comments**

*Reviewer: The authors present a scientific algorithm to simultaneously retrieve carbon monoxide and methane from TROPOMI onboard Sentinel-5 Precursor. I understand that this activity is important to calibration and validate both of TROPOMI spectra and operational products. However, it is unclear that what is the object for developing the proposed algorithm or how is the difference between the proposed algorithm and the operational one. In this paper, a lot of demonstrated results are described. Unfortunately, it is hard to understand the usefulness of this algorithm. It is easy to understand, if this paper focus on the validation of TROPOMI operational products. Several topics are described in this paper but the relevance between these topics are poor. Of course, the individual topics are important. So, I recommend the authors will reconstruct the frame of this paper.*

**Authors:** We have added details of the objectives of this new algorithm to the abstract and the introduction. The differences to the operational algorithm are described in Section 4.1. The goal of the paper is to introduce the TROPOMI/WFMD algorithm including error assessments based on synthetic data and validation with independent reference data to show that the algorithm is suitable to retrieve XCO and $XCH_4$ from real TROPOMI data well within the mission requirements after quality filtering. The good global agreement of our scientific products with the operational products for the analysed example cases further underlines the quality of the presented algorithm. The possibility to learn from the differences in detail is one of the advantages of having several distinct retrieval algorithms for each analysed atmospheric constituent at hand. Perhaps the most striking new feature of TROPOMI is the capability to readily detect emission sources in a single satellite overpass due to its unique combination of high precision, spatiotemporal resolution, and coverage. This enables new application areas and has the potential to advance emission monitoring and air quality assessments to an entirely new level because of the daily recurrence.

The leitmotif of Section 4 is to present initial results concerning comparison to the operational products and detection of emission sources. However, it is not the intention of the manuscript to give final answers in these areas, but rather to describe the objectives and outline the future potential as described in the conclusions. The comparison to the operational products is limited to example cases to demonstrate the broad consistency of the algorithms and the emission source analysis focusses on qualitative examples to demonstrate the new capabilities. A complete verification of the operational algorithms and the detailed quantitative reinforcement of the analysis of specific emission sources is out of the scope of this manuscript and will be discussed elsewhere. Along these lines, we have changed the name of Section 4 to "Initial

results using real TROPOMI data". We have also modified the abstract and the introduction to make this more clear. Please, see also the answers to the specific comments for more details.

**Specific comments**

**Abstract**

***Reviewer:*** *Describe the character of proposed science algorithm. Especially, the comparison between the proposed algorithm and the operational one.*
*Describe the motivation or object for developing proposed algorithm.*

**Authors:** The objectives have been added to the abstract in the revised version including the intention of mutual verification of the operational and the presented scientific algorithms.

***Reviewer:*** *Describe a full word of "DOAS".*

**Authors:** Has been added.

***Reviewer:*** *Page 2, line10: What is a "reference data".*

**Authors:** Has been changed to "validation data" for the sake of clarity.

***Reviewer:*** *Page 2, line13: Why emission sources have to be identified? Describe the object or background.*

**Authors:** We have added a sentence that the detection of emission sources in a single satellite overpass has the potential to advance emission monitoring and air quality assessments to an entirely new level.

***Reviewer:*** *I understand that one of target for developing this scientific algorithm is to validate the operational TROPOMI XCH4 and XCO products. If so, it might be described in.*

**Authors:** We have added a sentence that *mutual verification* of the operational and the presented scientific algorithms is one of the objectives. We think that the term *verification* is more suitable than *validation* when comparing satellite data sets because satellite data should not be considered as ground truth which is needed for validation.

**Introduction**

***Reviewer:*** *Page 4, line 9 to line 15: It is unclear what is the requirement of a scientific algorithm? I understand that validation of operational products, calibration of TROPOMI spectra, and reduce the random and systematic error of XCH4 and XCO with scientific algorithm are first objects. Second is new findings with scientific algorithm. If so, describe more clearly.*

**Authors:** Details of the objectives of this new algorithm are now described more clearly in the last paragraph of the introduction, which now reads: "Here we introduce a scientific algorithm to retrieve CO and CH$_4$ simultaneously from TROPOMI that has the objective

to complement the operational algorithms in the sense described above and to provide new geophysical insights, whilst performing within the mission requirements concerning random and systematic errors at the same time. The presented scientific algorithm differs from the operational algorithms in several respects (Landgraf et al., 2016; Hu et al., 2016) (see also Section 4.1 for a summary of the differences) and the corresponding products are thus pre-destined to be used together with the operational products in an ensemble approach. After a thorough description of the algorithm including error characteristics based on synthetic data and validation with independent reference data, we present first results of our new algorithm for both trace gases demonstrating the broad consistency with the operational products for example cases and the potential to advance the new application fields, for which TROPOMI's groundbreaking features pave the way."

**WFM-DOAS retrieval algorithm**

**Reviewer:** *Figure 1, It is unclear the coverage of gray hatch. Add table for these coverages.*

**Authors:** We have added the extent of the fitting windows to the caption of Figure 1 in the revised version.

**Reviewer:** *Table 2, Describe the meaning of "T", "p", "BL", "R", "$\tau$".*

**Authors:** We have changed $T$ to temperature, $p$ to pressure, and BL to *boundary layer* in Table 2 for a better understanding. The cloud optical thickness $\tau$ and the effective radius $R$ are now explained in the caption.

**Reviewer:** *Figure 5, Describe the full word of "cum", "cir"*

**Authors:** The abbreviations for water clouds (cumulus) and ice clouds (cirrus) are explained in the figure caption in the revised version.

**Reviewer:** *Figure 6, Describe the meaning of numbers for surface type.*

**Authors:** The meaning of the labels has been added to the figure caption in the revised version: 1 Crops, Mixed Farming; 2 Short Grass; 3 Evergreen Needleleaf Trees; 4 Deciduous Needleleaf Trees; 5 Deciduous Broadleaf Trees; 6 Evergreen Broadleaf Trees; 7 Tall Grass; 8 Desert; 9 Tundra; 10 Irrigated Crops; 11 Semidesert; 12 Ice Caps and Glaciers; 13 Bogs and Marshes; 14 Inland Water; 15 Ocean; 16 Evergreen Shrubs; 17 Deciduous Shrubs; 18 Mixed Forest; 19 Forest/Field Mosaic; 20 Water and Land Mixtures.

**Reviewer:** *Page 12, line 8 to 15: In the other algorithm to retrieving the XCH4 and XCO used O2A spectra, to identify the photon path with precisely. However, this algorithm is not employed the O2A spectra to identify the photon path. Instead of O2A spectra, this algorithm used the ECMWF-Based mole fraction computation. The authors are concluded that the proposed algorithm is more faster and accurate than that of O2A based processing system. However, it is not quantitative. The authors have to assess more quantitatively.*

**Authors:** The respective text passage has been changed accordingly and now reads "... For these reasons, $O_2$ is a barely sufficient proxy for the lightpath in the $2.3\,\mu$m spectral

range in a scattering atmosphere. For example, the $O_2$ errors for the scattering scenarios *aerosols/extreme in boundary layer* and *clouds/cirrus* from Table 2 are $-5.40\%$ and $-7.54\%$, respectively. Hence, the $O_2$ underestimations are considerably larger than the corresponding errors for $CH_4$ and $CO$, which would lead to distinct overestimations of mole fractions obtained from the $O_2$-proxy approach in the presence of strong scatterers.

In addition to the better accuracy of the ECMWF-based mole fraction computation, this approach is also faster, because the oxygen fit and the interband coregistration mapping can be omitted. As a consequence, the fitting procedure is about twice as fast without the normalisation by $O_2$. The ...”

**Reviewer:** *Page 13, line 22: Make table for "all 25 features".*

**Authors:** The 25 features are listed in the following sentence. This has been made more clear in the revised version.

**Reviewer:** *Page 15, Figure 8: What is the meaning of "QUAL=1"?*

**Authors:** QUAL=1 are excluded scenes of the implemented machine learning quality filter described in this section. We have added the explanation to the figure caption.

**Results**

**Reviewer:** *Page 24, line 4: Correct the capital position.*

**Authors:** Has been changed to "Shortwave Infrared CO Retrieval (SICOR)" in the revised version.

**Reviewer:** *Page 24, line 6: typo "amd".*

**Authors:** Has been changed.

**Reviewer:** *Page 25, Figure 15: Why the yield rate for XCH4 is drastically different between WFMD CO and Operational CO?*

**Authors:** As described in the main text describing Figure 15, the operational CO algorithm exhibits a better coverage as it can handle a larger amount of cloudiness.

**Reviewer:** *Page 27, Figure 18: How is the operational products? Is it possible to identify the emission sources with operational products?*

**Authors:** We have added a Figure showing the operational product and a related discussion in the revised version: "For comparison, Figure 18 also shows the operational product in addition to the TROPOMI/WFMD results. As the operational product is available as total CO columns, the corresponding mole fractions XCO were generated in the same way as for the scientific product by division of the total CO columns by the dry air columns obtained from ECMWF. The comparison demonstrates that the enhancements due to the analysed emission sources can be typically identified in both data sets. However, as a consequence of the different spatiotemporal sampling, the enhancement over some point sources is somewhat

more pronounced in the WFMD product. A possible reason for this is the additional utilisation of cloudy observations in the operational SICOR product, which may be associated with reduced surface sensitivity under certain conditions reflected in the averaging kernels of the corresponding measurements."

**Reviewer:** *Figure 19 to figure 22 are almost same information. These figures are just illustration. Make more clear sentence.*

**Authors:** These figures show examples of detection of emission sources in a single overpass for different regions, source types, source strengths, and trace gases. It is important to demonstrate that sufficiently large emission sources can be detected reliably. These various examples underpin that detection is the rule and not the exception. This enables new application areas like emission monitoring and air quality assessments as described in the introduction. A similar statement has also been added to the abstract in the revised version. See also the answer to the general comments.

**Conclusions**

**Reviewer:** *Page 33, line 10, The sentence "for example with respect to striping" is not touched on this paper. Adding the reference or explanation*

**Authors:** Striping in flight direction for single overpasses is introduced in Section 4.2.1 when discussing Figures 19 and 20. The meaning is made more clear in the conclusions of the revised version.